



# SAMM version 1.0: A numerical model for microbial mediated soil aggregate formation.

Moritz Laub[1,2], Sergey Blagodatsky[2,3], Marijn Van de Broek[1], Samuel Schlichenmaier[2],
Benjapon Kunlanit[5,6], Johan Six[1], Patma Vityakon[4,5], and Georg Cadisch[2]

[1]Sustainable Agroecosystems group, Department of Environmental Systems Science, Swiss Federal Institute of Technology (ETH Zurich), Universitätstrasse 2, 8092, Zurich, Switzerland
[2]Institute of Agricultural Sciences in the Tropics (Hans-Ruthenberg-Institute), University of Hohenheim, 70599 Stuttgart, Germany
[3]University of Cologne, Terrestrial Ecology Group, Institute of Zoology, Zülpicher Str. 47b, 50674 Cologne, Germany
[4]Department of Soil Science and Environment, Faculty of Agriculture, Khon Kaen University, Khon Kaen, Thailand
[5]Soil Organic Matter Management Research Group, Khon Kaen University, Khon Kaen, Thailand
[6]Department of Agricultural Technology, Faculty of Technology, Mahasarakham University, Maha Sarakham, Thailand

**Correspondence:** Moritz Laub (moritz.laub@usys.ethz.ch)

**Abstract.** In light of the large role that soil organic matter (SOM) plays in maintaining healthy and productive agricultural soils, it is crucial to understand the processes of SOM protection including the role of soil aggregate protection. Yet, few numerical process models include aggregate formation and even fewer represent the important connection between microbial growth and aggregate formation. Here, we propose a model of Soil Aggregation through Microbial Mediation (SAMM), which

consist of measurable pools and couples soil aggregate formation to microbial growth. The model was evaluated against data from a long term bare-fallow experiment in a tropical sandy soil, subject to plant litter additions of different compositions. The SAMM model effectively represented the microbial growth response after litter addition and the following formation and later disruption of aggregates. Model parameter correlation was low (all r < 0.5; r > 0.4 for only 4 of 22 parameters) showing that SAMM is well parameterized. Differences between treatments resulting from different litter compositions could be captured

by SAMM for soil organic carbon (Nash-Sutcliffe modelling efficiency (EF) of 0.68), microbial nitrogen (EF of 0.24) and litter carbon (EF of 0.80). Aggregate-related fractions, i.e., carbon inside aggregates (EF of 0.60) and also carbon in the free silt and clay fraction (EF of 0.24) were simulated very well to satisfactory. Analysis of model parameters led to further noteworthy insights. For example, model results suggested that up to 50% of carbon in the soil is stabilized through aggregate protection, even in a sandy soil, and that both microbial activity and physical aggregate formation coexist. When aggregate formation was

deactivated, the model failed to stabilize soil organic carbon (EF dropped to -3.68) and microbial nitrogen was represented less well (EF of 0.13). By re-calibrating the model version with deactivated aggregates, it was possible to partly correct for removing the aggregate formation, i.e., by reducing the decomposition rate of mineral attached carbon by about 85% (EF of 0.68, 0.75 and 0.18 for SOC, litter carbon and microbial nitrogen, respectively). Yet, the overall slightly better evaluation statistics (e.g., Akaike information critereon of 5351 vs 5554) show the potential importance of representing aggregate dynamics within SOM

models. Our results indicate that current models without aggregate formation partly compensate the missing protection effect



by lowering turnover rates of other pools and thus may still be suitable options where data on aggregate associated carbon is not available.

## 1 Introduction

Soil aggregates play a crucial role in the context of soil carbon sequestration because soil organic matter (SOM) stabilized in aggregates is a fraction of SOM that is strongly affected by human activities (Six and Paustian, 2014). There is evidence that the particulate organic matter (POM) stored within aggregates may be the SOM fraction that does least saturate if carbon inputs are increased (Castellano et al., 2015), and may thus be a suitable fraction to target for SOM accumulation. Yet, exactly this intra-aggregate POM becomes relatively easily available to decomposers upon disruption of aggregates (Six et al., 2000) and may therefore be considered to be labile. Mineral associated organic matter (MAOM), on the other hand, is thought as a part of SOM with slower turnover rates, but the pathways upon which it is formed are not completely clear. For example the concepts by Kallenbach et al. (2016) and Cotrufo et al. (2013) suggest that most stable MAOM is of microbial origin, whereas Angst et al. (2021) recently estimated that about half of MAOM is formed through direct adsorption of dissolved organic matter to soil minerals. As a result, we need a better understanding of the relative importance of the different processes of SOM stabilization, such as MAOM formation and POM protection within aggregates.

Numerical models are a good way to test our mechanistic understanding of complex systems, such as soils, and to improve knowledge about the interconnected processes by testing different hypotheses about the system. They allow to quantify fluxes which are not directly measurable and to test one or several conceptual structures of a system against measured data (Necpálová et al., 2015). Thus, they represent an elegant way to test research hypotheses. Despite the existence of conceptual models, the central role of microbial growth in aggregate formation is still incompletely understood and is only poorly represented in current SOM research models developed for the field scale. Initial attempts of Segoli et al. (2013), for example, modelled the formation and destruction of micro- and macroaggregates by including a simple microbial activity factor, but the model was not further developed into an ecosystem model and therefore is only applicable to shorter-term incubation experiments. The Millennial model (Abramoff et al., 2018, 2022) has a specific microbial biomass pool and distinguishes between aggregated and non-aggregated carbon, but its temporal dynamics have not been evaluated against long-term experiments and it does not simulate the effect of nitrogen on SOM dynamics.

In the sense of using models to test important research hypotheses, three important concepts/processes related to aggregate formation should therefore be included into models. The first important process to include into models of soil aggregate formation is the effect that plant residue composition and elemental stoichiometry (Sinsabaugh et al., 2013) have on carbon use efficiency (CUE) of microbes. For example, Lavallee et al. (2018) showed that shoot material leads to more stabilized MAOM than root material, which they attributed to a higher CUE for shoot material due to higher quality (i.e., low C/N and lignin; Cotrufo et al., 2013). Also Laub et al. (2022), in a long-term field experiment, found differences in aggregate dynamics between different litter type additions and suggested that these were a result of different CUE that depended on litter composition. Secondly, the effect of microbial activity on aggregate formation needs to be considered. Many studies in the





literature have shown the direct link between aggregate dynamics and microbial functioning. For example, Bucka et al. (2019)
showed - under incubation conditions - that microbial activity associated with dissolved organic matter and POM formed
aggregates rapidly. Thirdly, measurable pools. It has been suggested numerous times that next generation SOM models should
model carbon pools which are directly measurable (Segoli et al., 2013; Wang et al., 2013; Wieder et al., 2014). However, when
doing so one needs to adhere as much as possible to the principle of distinct structural identity (e.g. Oldfield et al., 2018; Wang
et al., 2022; de Aguiar et al., 2022). Thus within an optimal model based on measurable pools, any quantity of carbon should
maintain its structural identity until it is subject to an actual molecular change. This means that if carbon transfers from one
modeled pool to another, this should not only correspond to a transfer of matter between the pools, but also to a chemical or
physical reaction (e.g., depolymerization, anabolic microbial growth or adsorption to minerals). As such, MAOM and POM
have been identified as possible modelable pools of relative distinct structural identities (e.g. Segoli et al., 2013; Lavallee et al.,
2020) and are commonly accepted as the main building blocks for aggregates (Totsche et al., 2017). Furthermore, they can be
derived by established soil fractionation schemes and differ strongly in average turnover times and properties (Lavallee et al.,
2020; Schrumpf et al., 2013). It is, while POM consists mostly of undecomposed plant material, stabilized MAOM originates
either from microbial residues (Kallenbach et al., 2016; Six et al., 2006) or from dissolved organic matter (Angst et al., 2021).

Here, we present an approach to include all the above-mentioned concepts into a model of **Soil Aggregation** through
**Microbial Mediation** (SAMM). SAMM builds on the foundations introduced by mechanistic SOM models, such as simu-
lating measurable fractions and aggregates (Abramoff et al., 2018, 2022; Segoli et al., 2013) and the decomposition of plant
derived carbon to low molecular weight carbon, prior to consumption by microbes (Tang and Riley, 2015; Wang et al., 2013;
Zhang et al., 2021). It enriches these concepts by (i) the central role of microbes for soil aggregate formation and (ii) a consis-
tent structural identity of POM and MAOM within aggregates. We applied the model to simulate data from a long-term SOM
formation experiment in a tropical sandy soil in Northeast Thailand, which included inputs of litter of different compositions
and a non-amended control. SAMM is tested against measured data of microbial biomass, SOC and carbon in different soil
fractions. To better understand the model and it's uncertainty, a Bayesian calibration of model parameters is performed. The
calibrated model was then used to test three main hypotheses:

1. Simulating the connection between microbial growth and aggregate formation with SAMM helps to quantify the relative
   importance of different SOM stabilizing processes.

2. Including this connection into SOM models is essential to accurately represent dynamics of SOM formation. Thus, a
   model that explicitly simulates aggregate formation as a result of microbial growth will outperform a model of similar
   structure that does not include aggregate formation.

3. The dynamics of microbial activity, which are linked to temperature, moisture and litter composition, help to explain
   dynamics in aggregate formation. Thus, we expect that aggregates can be simulated with a similar model performance
as microbial biomass.





**Table 1.** Chemical characteristics of applied organic residues/litter. Total carbon was measured by Walkley and Black wet digestion; total nitrogen by micro-Kjeldahl, lignin and cellulose by acid detergent lignin method (Van Soest and Wine, 1968); polyphenols were determined according to Anderson and Ingram (1993). Values within the same column that share the same capital letter are not significantly different (p < 0.05). The table is adopted from Laub et al. (2022) under the creative common license 4: http://creativecommons.org/licenses/by/4.0/.

| Litter type (Abbreviation) | | Carbon (g kg⁻¹) | Nitrogen (g kg⁻¹) | C/N (g g⁻¹) | Lignin (g kg⁻¹) | Polyphenols (g kg⁻¹) | Cellulose (g kg⁻¹) |
|---|---|---|---|---|---|---|---|
| Rice straw | (RS) | 367$^A$ | 4.7$^A$ | 78$^A$ | 28.7$^A$ | 6.5$^A$ | 507$^A$ |
| Groundnut stover | (GN) | 388$^A$ | 22.8$^B$ | 17$^B$ | 67.6$^A$ | 12.9$^A$ | 178$^{AB}$ |
| Dipterocarp | (DP) | 453$^B$ | 5.7$^A$ | 80$^A$ | 175.5$^B$ | 64.9$^B$ | 306$^{AB}$ |
| Tamarind | (TM) | 427$^B$ | 13.6$^C$ | 32$^C$ | 87.7$^C$ | 31.5$^C$ | 143$^B$ |
| SE$^+$ | | 7 | 0.8 | 3.4 | 19 | 5.6 | 46 |

$^+$Standard error

## 2 Material and Methods

### 2.1 Description of the experiment

We tested the capability of SAMM in a long-term bare fallow experiment, which was established on a degraded tropical sandy soil in 1995 (Vityakon et al., 2000; Puttaso et al., 2011, 2013; Laub et al., 2022). In brief, the experiment was initiated to study the effects of annual additions of organic material (at a rate of 10 t dry matter ha⁻¹ yr⁻¹) of different composition on soil organic matter dynamics. The experiment is located within the research station of the Office of Agriculture and Cooperatives of the Northeast, Khon Kaen province (16°20' N; 102° 49' E) in Northeast Thailand. The soil is a Khorat sandy loam (Typic Kandiustult in USDA, Acrisol in WRB classification) with 90% sand and 5% clay (Puttaso et al., 2013). At the start of the experiment, the bulk density was 1.45 g cm⁻³, the pH was 5.5 and CEC 3.53 cmol kg⁻¹ in the 0-15 cm topsoil (Vityakon et al., 2000). Later measurements did not find significant changes in bulk density due to the treatments (data not shown), so we assumed a constant bulk density of 1.45 g cm⁻³ throughout the whole period for all treatments in this study. The site has a savanna type climate with a wet period from April to September with about 1200 mm annual precipitation and a mean temperature of 28°C (Puttaso et al., 2013). The experiment was a randomized complete block design with three replicated plots of 4 × 4 m size. The annual litter application of 10 t ha⁻¹ dry matter at the beginning of the rainy season around May, supplied about 4 t carbon ha⁻¹ yr⁻¹. Next to an unamended control (CT), the litter treatments were rice (*Oryza sativa*) straw (RS; high C/N, low lignin/polyphenol contents), groundnut (*Arachis hypogaea*) stover (GN; low C/N, low lignin/polyphenol contents), tamarind (*Tamarindus indica*) litter (TM; medium C/N, medium lignin/polyphenol contents) with leaf/petiole litter ratio of 7:1, and dipterocarp (*Dipterocarpus tuberculatus*; DP; high C/N, high lignin/polyphenol contents) leaf litter (Table 1). The applied litter was manually incorporated into the topsoil until a depth of approximately 15 to 20 cm using hand hoes. Hand weeding was conducted to keep plots vegetation free. This was done about once a month during the rainy season and every second month for the rest of the year, attempting to have as little as possible additional organic matter inputs from weeds.





**Table 2.** Overview of all measurements from the Khon Kaen long-term experiment that were used in this study.

| Type | Unit[*] | Frequency | Weeks[+] | Time span and reference |
|------|---------|-----------|----------|-------------------------|
| Litterbag C | kg C ha$^{-1}$ | 6 yr$^{-1}$ | 0, 2, 4, 8, 16, 32 | 2004[a] |
| Microbial N | kg N ha$^{-1}$ | 6 yr$^{-1}$ | 0, 2, 4, 8, 16, 32 | 1995[b], 96-99[X], 2004[a], 07[X], 12[X], 19[c] |
| Soil organic C | kg C ha$^{-1}$ | 1 yr$^{-1}$ | 0 | 1995-2005[d], 2006-16[X], 2019[c] |
| Soil C/N | g g$^{-1}$ | 1 yr$^{-1}$ | 0 | 1995-2005[d], 2006-16[X], 2019[c] |
| Aggregate C | kg C ha$^{-1}$ | 6 yr$^{-1}$ | 0, 2, 4, 8, 16, 30 | 2019[c] |
| Free mineral associated C | kg C ha$^{-1}$ | 6 yr$^{-1}$ | 0, 2, 4, 8, 16, 30 | 2019[c] |

[*]Data rescaled to kg ha$^{-1}$ using 20 cm soil depth and a bulk density of 1.45 g cm$^{-3}$; [+]Weeks after residue addition (0 = prior); References: [a]Puttaso et al. (2011), [b]Vityakon et al. (2000), [c]Laub et al. (2022), [d]Vityakon (2007), [X]Unpublished

However, despite best efforts it was not possible, to keep the plots completely free of vegetation at all times. The experimental data covered a time period from establishment of the experiment in 1995 until December 2019.

## 2.2 Measurements available from the long-term experiment

Soil microbial biomass carbon and nitrogen data were available from most years and always measured prior to litter incorporation and in weeks 2, 4, 8, 16 and 32 after litter addition (Puttaso et al., 2011; Vityakon et al., 2000; Vityakon, 2007; Laub et al., 2022, and unpublished data in Table 2). Litterbag decomposition experiments were conducted to elucidate differences in litter decomposition rates as a function of litter composition, measuring ash-free dry weight remaining at the same points in time (Puttaso et al., 2011). Soil microbial biomass was measured by chloroform fumigation extraction (see Puttaso et al., 2011, for

more details). Because microbial carbon and nitrogen are usually correlated, we only made use of the microbial nitrogen data, which was of higher quality (fewer negative values than carbon, lower variability within treatments). Annual measurements of soil organic carbon and soil C/N data, measured by Walkley-Black method (Walkley and Black, 1934), were available from Vityakon et al. (2000) and from further annual measurements until 2016 and from 2019. Additionally, there were measurements of carbon in aggregates (carbon in small macroaggregates, 2–0.25 mm; and microaggregates, 0.25–0.053 mm; combined) and

the free silt and clay faction (MAOC) throughout the year 2019 at weeks 0, 2, 4, 8, 16 and 30 (Laub et al., 2022).

## 2.3 The SAMM model version 1.0: Core concepts and model description

The core concepts of SAMM are 1) all pools are measurable entities that have a conceptual carbon identity (Wang et al., 2022), which they maintain inside aggregates and along the gradient of increased decomposition status, 2) linking aggregate formation to the microbial life cycle and 3) simulating aggregates in a coupled soil carbon and nitrogen model. For brevity, we

only explain the central concepts of SAMM and the flow of carbon and nitrogen in the main text, while the appendix hosts a detailed description of model pools (A1) and the differential equations comprising the SAMM model (A3). A list of all model pools is given in Table 3, while all parameters and their calibrated values are given in Table 4.



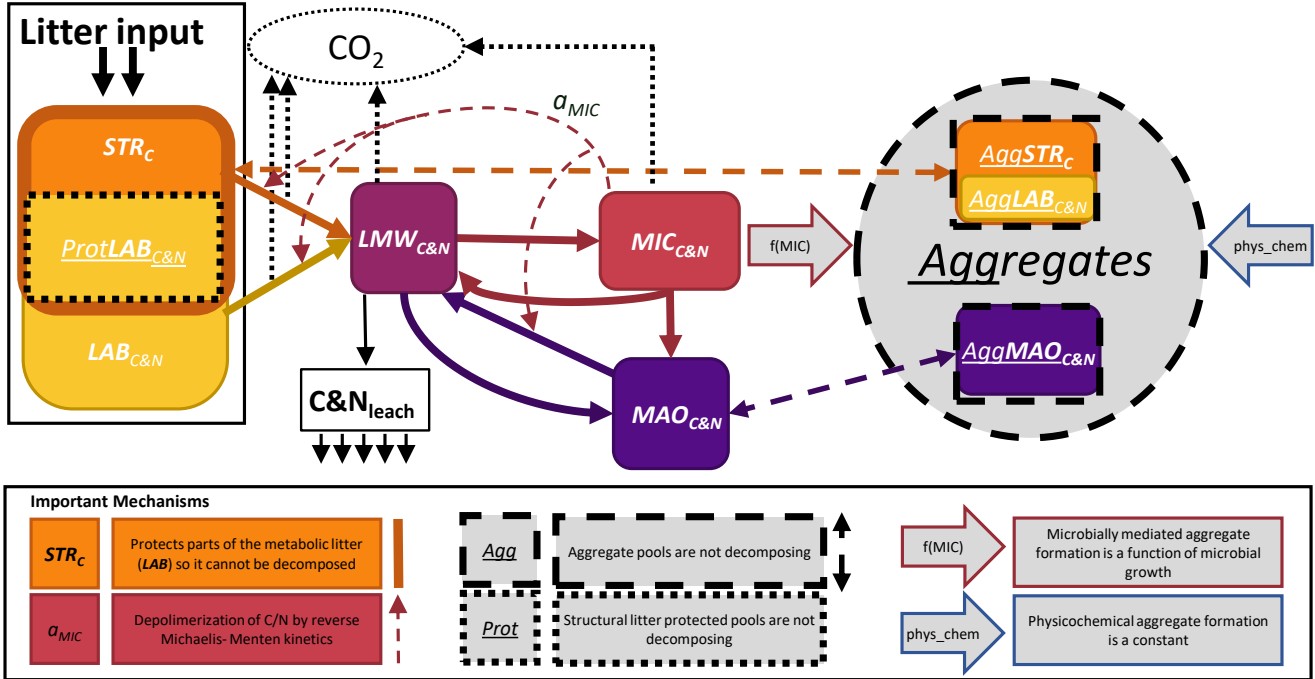

**Figure 1.** Conceptual model of SAMM. Carbon and nitrogen in pools are depicted as $Pool_{C\&N}$ or $Pool_C$ for carbon only pools. The following pools exist: $STR_C$, structural litter; $LAB_{C\&N}$, labile litter; $LMW_{C\&N}$, low molecular weight; $MIC_{C\&N}$, microbial; $MAO_{C\&N}$, mineral associated. Thick continuous arrows represent flows of carbon and nitrogen between pools which include a change in structural identity. Thick, dashed arrows represent aggregate protection and deprotection, which does not change the structural identity. The effect of $MIC_{C\&N}$ on pool decomposition by reverse Michaelis Menten kinetics ($a_{MIC}$ paramter) is represented by the thin dashed arrow. The two large arrows with coloured outline represent the factors that influence the rate of aggregate formation. Losses from the system are depicted by thin dotted ($CO_2$) and continued arrows (leaching). Further abbreviations: *Prot*, protected by structural litter; *Agg*, aggregate protected pools.

To achieve full measurability, simulated fresh litter was divided into two pools, structural litter measured as lignin and polyphenols (similar to Campbell et al., 2016), and metabolic (labile) litter representing the remaining litter carbon and nitro-

gen, thus enabling different CUE and decomposition rates resulting in differences in microbial growth. Through simulating both carbon and nitrogen, the model further allows for a C/N ratio-dependent CUE at microbial uptake. Carbon and nitrogen cycles are coupled (Fig. 1 and Table 3), but the structural litter pool is defined as a carbon-only pool. This is indicated in the following by the subscripts next to the pool names (i.e., $POOL_C$ for carbon only, and $POOL_{C\&N}$ for carbon and nitrogen containing pools).

The organic matter decomposition process within the SAMM model starts with undecomposed plant material, consisting of structural litter ($STR_C$), and the metabolic/labile litter pool ($LAB_{C\&N}$). Upon depolymerization, the carbon and nitrogen of any pool enters the easily soluble low molecular weight ($LMW_{C\&N}$) pool. This $LMW_{C\&N}$ is the only pool that contains





**Table 3.** An overview of all SAMM model pools and their units.

| Pool | Description | Unit[+] |
|---|---|---|
| $STR_C$ | Structural litter pool C | kg C ha$^{-1}$ |
| $LAB_C$ | Metabolic litter pool C | kg C ha$^{-1}$ |
| $LAB_N$ | Metabolic litter pool N | kg N ha$^{-1}$ |
| $LMW_C$ | Low molecular weight pool C | kg C ha$^{-1}$ |
| $LMW_N$ | Low molecular weight pool N | kg N ha$^{-1}$ |
| $MIC_C$ | Microbial biomass pool C | kg C ha$^{-1}$ |
| $MIC_N$ | Microbial biomass pool N | kg N ha$^{-1}$ |
| $MAO_C$ | Mineral associated C | kg C ha$^{-1}$ |
| $MAO_N$ | Mineral associated N | kg N ha$^{-1}$ |
| $AggSTR_C$ | Structural litter pool C protected in aggregates | kg C ha$^{-1}$ |
| $AggLAB_C$ | Metabolic litter pool C protected in aggregates | kg C ha$^{-1}$ |
| $AggLAB_N$ | Metabolic litter pool N protected in aggregates | kg N ha$^{-1}$ |
| $AggMAO_C$ | Mineral associated C protected in aggregates | kg C ha$^{-1}$ |
| $AggMAO_N$ | Mineral associated N protected in aggregates | kg N ha$^{-1}$ |

[+]For a defined depth interval (here 0 - 15 cm).

molecules that are small enough to be incorporated by the microbial biomass ($MIC_{C\&N}$). The production of extracellular enzymes consumes energy, which is indirectly accounted for by a pool-dependent carbon use efficiency (*CUE*), leading to the

respiration of CO$_2$ in the amount of *(1-CUE)* during the transition from any litter pool to $LMW_{C\&N}$. When the $MIC_{C\&N}$ pool consumes $LMW_{C\&N}$, a portion of the consumed carbon is respired as growth respiration, the rest is used for anabolism. The amount of growth respiration of $MIC_{C\&N}$ depends on a variable stoichiometric CUE, which is a function of the C/N ratio of $LMW_{C\&N}$. A fraction of $MIC_{C\&N}$ dies each time step and microbes also have a maintenance respiration. Part of it (the cell walls) are attached to minerals creating mineral associated carbon and nitrogen ($MAO_{C\&N}$), the rest (cell internal content)

is transferred back into the $LMW_{C\&N}$ pool. Direct adsorption of $LMW_{C\&N}$ to $MAO_{C\&N}$ is also possible. Carbon and nitrogen from the primary constituents (i.e., $LAB_{C\&N}$, $STR_C$, $MAO_{C\&N}$) get protected by integration into aggregates as a byproduct of microbial growth, i.e., the amount of aggregate formation is a function of microbial growth. There is also a physicochemical aggregate formation, which for simplicity is assumed to be constant in this version of SAMM. While inside the aggregates there is no decomposition, a concept proposed by Luo et al. (2017) as a way to reduce the number of parameters in aggregation

models. The carbon of all pools outside of aggregates is subject to decomposition by $MIC_{C\&N}$ following reverse Michaelis-Menten kinetics, a good approximation of enzymatic depolimerization (Abramoff et al., 2022; Tang and Riley, 2019). Thus, the speed of decomposition depends on the amount of substrate and the amount of $MIC_{C\&N}$. Aggregate disruption is simulated as a first order kinetic process.



**Table 4.** Overview of all SAMM model parameters (top), further computed helper variables (middle) and external model drivers and site conditions needed (bottom). The calibrated values are the best parameter set from the independent Bayesian calibration for the SAMM model and the recalibrated non aggregate model (SAMMnoAgg).

| Variable | Description | Units | Calibrated | SAMM[1] | SAMMnoAgg[2] |
|---|---|---|---|---|---|
| $k_{STR}$ | Turnover rate of structural litter pool | g g$^{-1}$ d$^{-1}$ | Yes | 0.0024 | 0.0028 |
| $k_{LAB}$ | Turnover rate of metabolic litter pool | g g$^{-1}$ d$^{-1}$ | Yes | 0.0225 | 0.0551 |
| $k_{MIC}$ | Death rate of microbial biomass pool | g g$^{-1}$ d$^{-1}$ | Yes | 0.0046 | 0.0098 |
| $k_{MAO}$ | Turnover rate of mineral associated carbon pool | g g$^{-1}$ d$^{-1}$ | Yes | 0.00044 | 0.000057 |
| $\mu_{max}$ | Maximum uptake rate of LMW by microbes | g g$^{-1}$ d$^{-1}$ | Yes | 0.238 | 0.367 |
| $k_{Agg}$ | Turnover rate of aggregate pools | g g$^{-1}$ d$^{-1}$ | Yes | 0.0316 | 1[x] |
| $K_{M_{MIC}}$ | Half-saturation constant of the microbial activity factor | - | Yes | 35.5 | 1.0 |
| $m_{MIC}$ | Maintenance respiration of microbes | g g$^{-1}$ d$^{-1}$ | Yes | 0.00035 | 0.0013 |
| $K_{LMWMAO}$ | Specific adsorption rate of LMW to MAOM | g g$^{-1}$ d$^{-1}$ | Yes | 0.043 | 0.031 |
| $c_{SORP}$ | Maximum sorption capacity coefficient | g g$^{-1}$ | No[*] | 0.83 | 0.83 |
| $CUE_{STR}$ | Carbon use efficiency of structural litter pool | g g$^{-1}$ | Yes | 0.65 | 0.52 |
| $CUE_{LAB}$ | Carbon use efficiency of metabolic litter pool | g g$^{-1}$ | Yes | 0.54 | 1.00 |
| $CUE_{LMW}$ | Maximum carbon use efficiency of low molecular weight pool | g g$^{-1}$ | No[+] | 0.6 | 0.6 |
| $CN_{min(MIC)}$ | Minimum C/N ratio of microbial biomass pool | g g$^{-1}$ | Yes | 5.01 | 6.12 |
| $CN_{max(MIC)}$ | Maximum C/N ratio of microbial biomass pool | g g$^{-1}$ | Yes | 10.1 | 9.49 |
| $f_{MICMAOM}$ | Fraction of MIC directed to MAOM upon microbial death | g g$^{-1}$ | Yes | 0.24 | 0.26 |
| $pc_{STR_{LAB}}$ | Protection capacity of $STR_C$ for $LAB_{C\&N}$ | g g$^{-1}$ | Yes | 2.47 | 3.98 |
| $aggfactSTR_C$ | Protection of $STR_C$ inside aggregates per microbial growth | g g$^{-1}$ | Yes | 0.71 | 0[x] |
| $aggfactMAO_C$ | Protection of $MAO_C$ inside aggregates per microbial growth | g g$^{-1}$ | Yes | 2.70 | 0[x] |
| $NonMicAgg$ | Physicochemical aggregate formation | kg $MIC_Ceq$ ha$^{-1}$ d$^{-1}$ | Yes | 31.0 | 0[x] |
| $DailyLitter_C$ | Daily root carbon inputs (from unavoidable plant growth) | kg C ha$^{-1}$ d$^{-1}$ | Yes | 3.07 | 3.09 |
| $DailyLitter_{C/N}$ | C/N ratio of daily root inputs | g g$^{-1}$ | Yes | 159.3 | 47.0 |
| $DailyLitter_{STRC(\%)}$ | Percent of strucural litter in daily root inputs | g g$^{-1}$ | Yes | 0.13 | 0.24 |
| **Computed helper variables (rate modifiers etc.)** | | | | | |
| $CUE_{CN(LMW)}$ | Dynamic C/N based carbon use efficiency of $LMW_C$ pool | g g$^{-1}$ | - | - | - |
| $s_t$ | Temperature scalar | - | - | - | - |
| $s_w$ | Water scalar | - | - | - | - |
| $pLAB$ | Fraction of metabolic litter protected by structural litter | g g$^{-1}$ | - | - | - |
| $a_{MIC}$ | Michaelis-Menten microbial activity factor | - | - | - | - |
| $MAO_{C_{max}}$ | Maximum adsorption capacity to $MAO_C$ | t ha$^{-1}$ | - | - | - |
| $w_{leach}$ | Share of soil water leached (HYDRUS calculation) | g g$^{-1}$ d$^{-1}$ | - | - | - |
| **Site condition and other model driving variables** | | | | | |
| $depth$ | Soil depth to be simulated | m | - | - | - |
| $BD$ | Bulk density | kg m$^{-3}$ | - | - | - |
| $\%SiCl$ | Silt and Clay fraction | % | - | - | - |

[1]Model version including soil aggregates; [2]Recalibrated model version without soil aggregates; [*]from Abramoff et al. (2022); [+]established maximum (Sinsabaugh et al., 2013; Manzoni et al., 2012); [x] set to 0/1 in model version without soil aggregates to deactivate them.



## 2.4 SAMM setup and Bayesian calibration

For the technical implementation of SAMM version 1.0, we used the R programming language (R Core Team, 2020). The details are described in appendix A2. As SAMM is a new model, most model parameters needed to be calibrated. In addition to typical SOM model parameters representing pool turnover, SAMM contains some unique parameters, such as the protection capacity that $STR_C$ exhibits on $LAB_{C\&N}$, the rate of aggregate formation per microbial growth, and the rate of physicochemical aggregate formation (Table 4). Also, the amount and composition of carbon and nitrogen entering the soil via plant roots were

calibrated parameters. These were necessary because, despite best attempts to keep the experiment completely fallow, it was not possible to completely eliminate plant growth in the plots. Two model parameters were fixed based on literature. The first uncalibrated parameter was the maximum CUE for $LMW_C$, which was fixed to 0.6 (Sinsabaugh et al., 2013; Manzoni et al., 2012). The second uncalibrated parameter was $c_{SORP}$, the maximum sorption capacity of the fine fraction, which was taken from Abramoff et al. (2022).

To test our hypotheses about the importance of aggregates in carbon stabilization and the need to simulate this process, we also created a SAMM version without aggregate formation (SAMMnoAgg). By setting the turnover of aggregates ($k_{Agg}$) to 1 $d^{-1}$ and the aggregate formation parameters to 0, all aggregate protection was effectively removed from the model. We assessed the difference in simulated stabilized SOC in SAMM and SAMMnoAGG, using the parameters calibrated for SAMM, to gain insights into the importance of aggregate protection for SOC stabilization. SAMMnoAGG was further recalibrated to test our

hypothesis of the need to simulate aggregates to represent SOM dynamics. Note that measurements of carbon in aggregates and in the silt and clay fraction from 2019 were not used in recalibrating SAMMnoAGG.

As a starting point for model parameters, an initial model calibration was performed using a genetic algorithm (GA package of R; Scrucca, 2013). To explore the uncertainty associated with the two different versions (i.e., SAMM and SAMMnoAgg), this initial calibration was followed by a Bayesian calibration applying the sampling importance resampling (SIR) method. This

method was used by Gurung et al. (2020) to calibrate the SOM module of DayCent and is described in detail in their article. Briefly, the SIR method uses Bayes' theorem to derive the posterior distribution of model parameters and model outputs based on an assumed prior and available data. We assumed normally distributed broad priors centered around the initial calibrated model parameters, i.e., the mean parameter values from SAMM and SAMMnoAgg to have the same priors for both (except for the values only calibrated in the aggregate version). In the next step of SIR, the posteriors are derived by filtering the

prior using importance weights to sample individual parameter sets from the prior. The importance weights are proportional to the simulation likelihoods (i.e., of observing the data, given the model), which are computed using the data, the simulated values and the variance-covariance matrices of data (Wallach et al., 2019). As is common practice, we assumed that the covariances were zero, hence we only used the variances for each type of measurement (taking the median variance computed for each type of data from the three experimental repetitions). Then, by dividing the likelihood of each simulation by the mean

likelihood of all simulations, standardized importance weights were computed. The prior parameter set was then resampled without replacement and the importance weights taken as sampling probability. Overall, a total of 200.000 simulations were performed, of which 200 parameter sets were drawn in the resampling.





## 2.5 Model evaluation

The following standard evaluation statistics were used for model evaluation, as defined by Loague and Green (1991):

$$MSE_y = \frac{1}{n}\sum_{z=1}^{n}(O_{yz} - P_{yz})^2 \tag{1}$$

$$RMSE_y = \sqrt{MSE_y} \tag{2}$$

$$EF_y = 1 - \frac{\sum_{z=1}^{n}(O_{yz} - P_{yz})^2}{\sum_{z=1}^{n}(O_{yz} - \bar{O}_y)^2} \tag{3}$$

Here, $MSE_y$ is the mean-squared-error and $RMSE$ is its root. $EF_y$ is Nash-Sutcliffe modelling efficiency, $O_{yz}$ stands for the measured value of the *z-th* measurement of the *y-th* type of measurement. Further, $\bar{O}_y$ is the mean of measured values of the *y-th* type of measurement and $P_{yz}$ is the model predicted value corresponding to $O_{yz}$. As suggested by Gauch et al. (2003) to gain a better insight into the nature of model errors, we further divided $MSE_y$ into the squared bias (*SB*), nonunity slope (*NU*) and lack of correlation (*LC*). We expressed them in relative terms, by dividing them by the $MSE_y$:

$$SB_y(\%) = \frac{(\bar{O}_y - \bar{P}_y)^2}{MSE_y} * 100 \tag{4}$$

$$NU_y(\%) = \frac{(1 - b_y)^2 * (\frac{\sum_{z=1}^{n}(O_{yz}^2)}{n})}{MSE_y} * 100 \tag{5}$$

$$LC_y(\%) = \frac{(1 - r_y)^2 * (\frac{\sum_{z=1}^{n}(P_{yz}^2)}{n})}{MSE_y} * 100 \tag{6}$$

Here, $\bar{P}_y$ is the mean predicted value of the *y-th* measurement type, *b* the slope of the regression of *P* on *O*. Finally, *r* is the correlation coefficient between *O* and *P*. The relative *LC*, *SB* and *NU* provide information if the model errors are mostly random (high *LC*) or whether there is a systematic bias (high *SB*). A high relative *NU* indicates that the model sensitivity is wrong (either too low or too high). The *SB* can be interpreted as the intercept of a regression between predictions and observations, whereas the *NU* is the slope of this regression (Gauch et al., 2003). Finally, the Akaike information criterion (AIC) was computed to compare different model versions:

$$AIC = 2k - 2ln(\bar{L}) \tag{7}$$

Here, *k* is the number of model parameters that were estimated (23 for SAMM and 19 SAMMnoAgg) and $\bar{L}$ is the likelihood.





## 3   Results

Because SAMM is a new model, we first describe its behaviour and illustrate the development of pools (Fig. 2) using the
treatment with the highest microbial activity, the groundnut stover treatment. It is important to note that our results cover a time
period where the model has not yet reached a new steady state. Second, the performance of the calibrated model is evaluated
against the measured data and posterior parameter distributions are discussed. Third, we test the importance of aggregate
protection in SAMM, by assessing how much the simulation performance decreases for different types of measurements when
aggregate formation is not simulated (SAMMnoAGG). Finally, we try to assess to which extent simulating aggregate formation
is necessary to correctly simulate microbial biomass and SOC, by recalibrating the SAMMnoAGG version and comparing it
to SAMM.

### 3.1   SAMM model behavior: the connection between microbes and aggregate formation

After the groundnut stover application in the year 2001, a rapid depolymerization of the part of $LAB_C$ that is not protected by
$STR_C$ is simulated (Fig. 2). The depolymerized material is transferred to the $LMW_C$ pool. This increase in $LMW_C$ feeds the
growth of $MIC_C$, which almost triples in biomass. The $MIC_C$ growth slows down once the unprotected part of $LAB_C$ is fully
decomposed. Yet, the peak of $LMW_C$ availability is within one to two weeks after litter addition, while the peak of $MIC_C$ is about
one to two months after litter addition and maximum $LMW_C$ availability. The increase of microbial growth is accompanied by
an increase in the formation of new aggregate-protected carbon. Unprotected $MAO_C$ and litter get thereby protected in the
aggregates, increasing the amount of aggregate protected $MAO_C$ and litter by about 30%. Because the formation of aggregates
is linked to microbial growth, the peak of aggregate protected pools ($MAO_C$, $LAB_C$ and $STR_C$) occurs simultaneously with
the peak of $MIC_C$. Thereafter, the amount of aggregate carbon starts to reduce again, which becomes visible in the increase
of unprotected $MAO_C$, $LAB_C$ and $STR_C$. During the dry season about 250 days after residue application, another increase
in aggregate formation occurs, this time driven by the physicochemical aggregate formation that continues while aggregate
turnover is reduced due to limiting water availability. After a full year, just prior to the next addition of litter, most of the newly
added litter of the year before is decomposed and increased moisture availability increases aggregate disruption again. Yet, a
higher amount of $MAO_C$ compared to the beginning of the year, and a slightly higher amount of aggregate protected $MAO_C$,
$STR_C$ and $LAB_C$ leads to an increased amount of SOC compared to the previous year.

### 3.2   Evaluating SAMM against measured data

Overall, the SAMM model was capable of simulating the different types of available measurements, as indicated by positive
modelling efficiencies for all of them (Table 5a; soil C/N was the only exception). The best representation of measured values by
the model was that of residue-C in litterbags (Fig. 3; EF 0.80) and interestingly, the measured groundnut stover decomposition
was so fast (>50% in the first week) that the model could not capture it. Also the measured values of topsoil SOC were
represented well by SAMM (Fig. 4 and 5; EF 0.68), with a tendency of the model to overestimate SOC in the rice straw
treatment. Further, microbial nitrogen ($MIC_N$; EF 0.24) and carbon in the free silt and clay fraction ($MOA_C$; EF 0.24) were





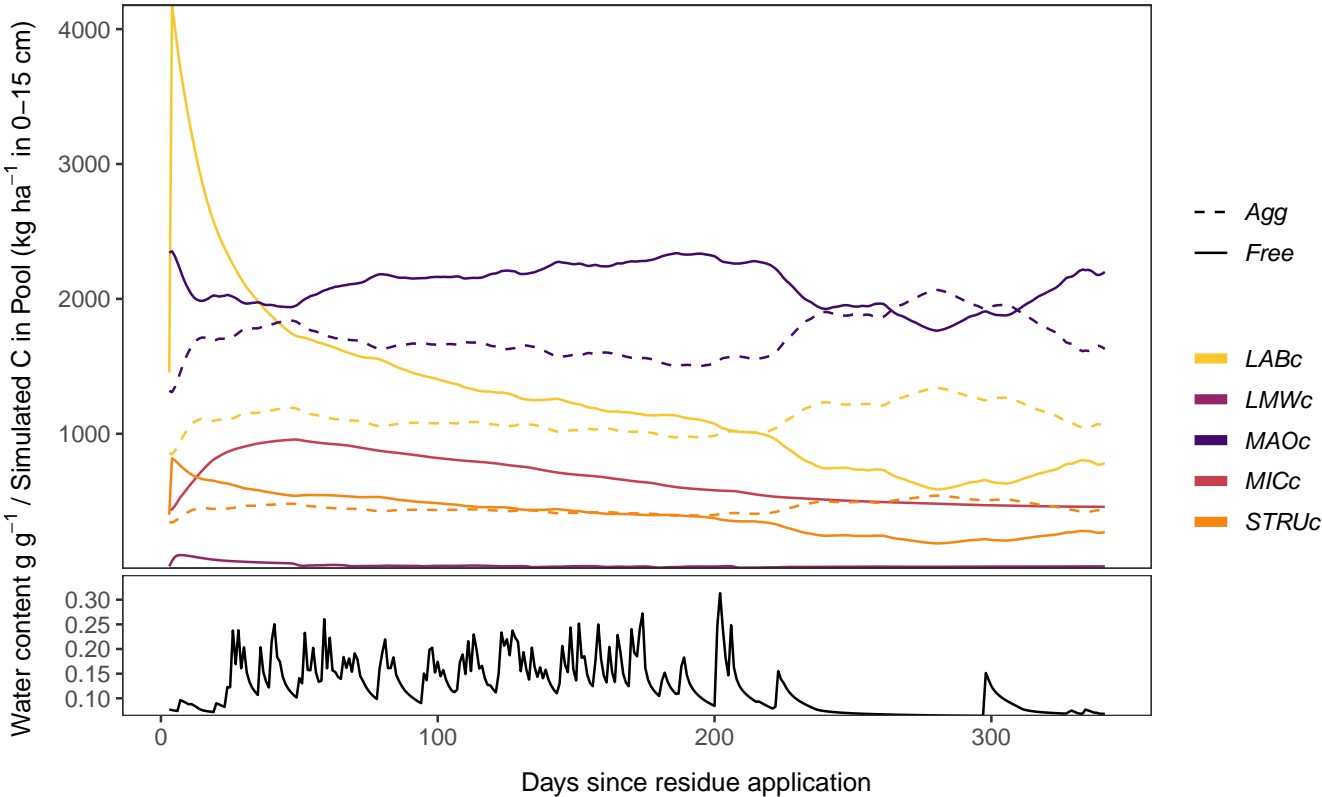

**Figure 2.** Exemplary SAMM model behavior amd pooly carbon dynamics of the groundnut treatment in the year 2001 to 2002 starting a day before the addition of litter. The top figure displays all carbon pools inside and outside of aggregates, while the bottom figure displays the soil water content (model driver, simulated by HYDRUS 1D). In the two figure, aggregate protected pools (Agg) are represented by a dashed line, decomposable (Free) pools by a solid line. $STR_C$, structural litter; $LAB_C$, labile litter; $LMW_C$, low molecular weight; $MIC_C$, microbial; $MAO_C$, mineral associated.

simulated with acceptable accuracy (Fig. 5 and 6). The temporal trend of microbial nitrogen was also captured well for all litter treatments with the exception of the control, in which there was almost no simulated microbial growth response over the year (Fig. 6). For free ($MOA_C$, the differences between treatments were captured, and the temporal dynamic was low, both in measured and modelled values. The temporal variation of free $MAO_C$ was minimal both in measurements and simulated values and the model could overall capture the treatment differences (EF 0.24). It could also very well capture the temporal dynamics of aggregate C in the groundnut, rice straw and tamarind treatments, as well as the absence of major temporal dynamics in the other two treatments (Fig. 6; EF 0.60). Despite the dynamic CUE function of SAMM, the SOC content of the high C/N ratio residue treatments (rice straw most strongly and dipterocarp to some extent) tended to be overestimated while tamarind tended to be underestimated, leading to poor model performance (EF -0.58; Fig. 5).





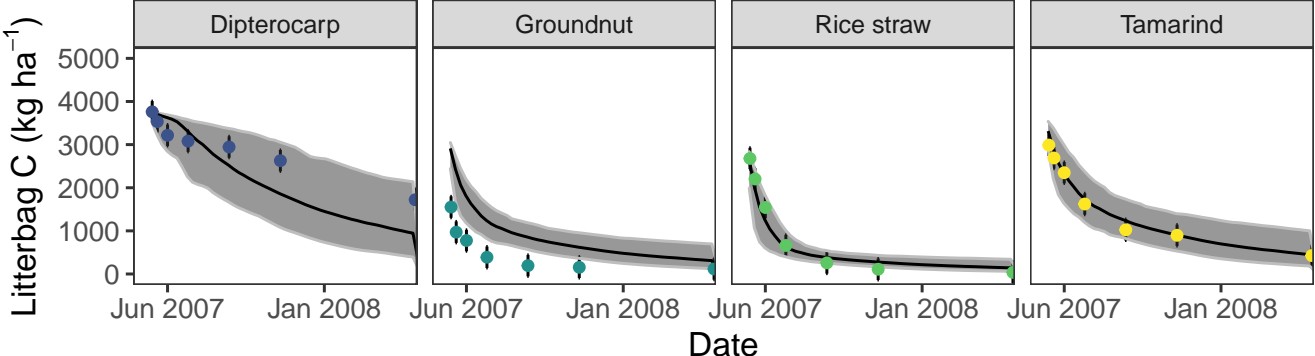

**Figure 3.** Simulation of incubated litterbag residue-C dynamics from different litter materials (burried at 15 cm depth). Dots with error bars indicate the mean and 95% credibility interval of observations. The black line and grey band indicate the best simulation and the 95% credibility interval of the Bayesian calibration posterior, respectively.

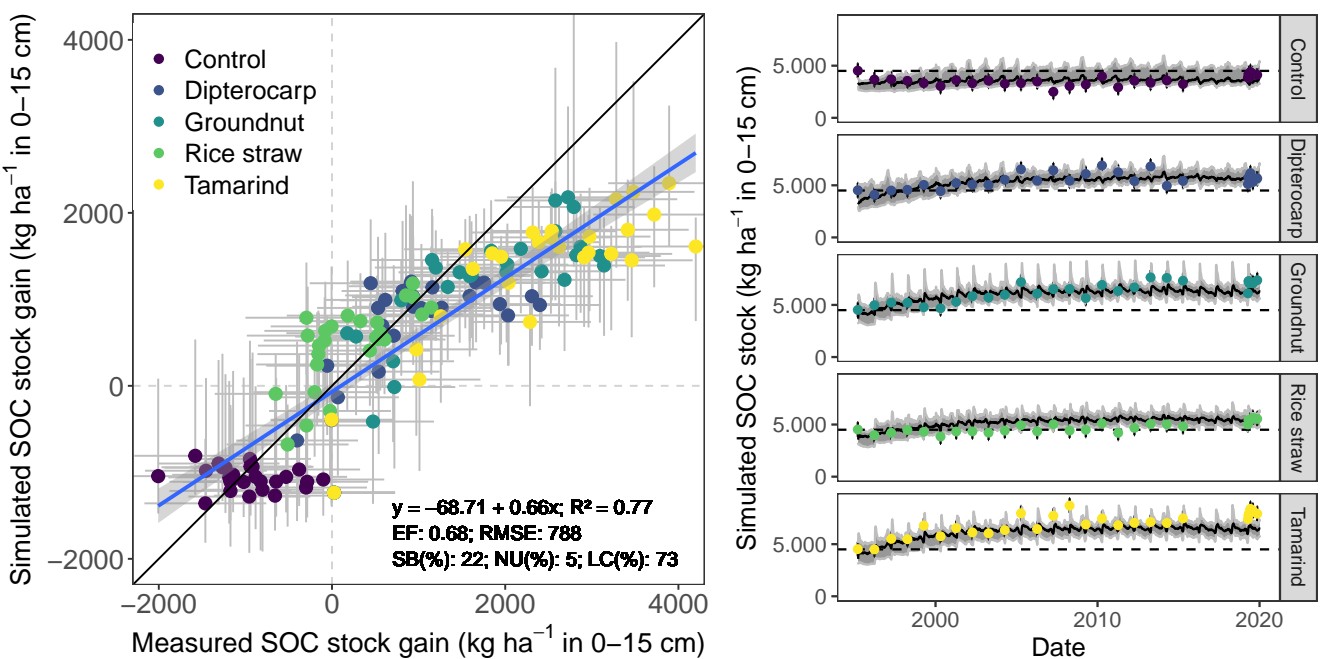

**Figure 4.** Measured and simulated development of SOC stocks in the top 15 cm of soil from all residue addition treatments. Displayed are the measured versus modelled gain in SOC stocks since the onset of the experiment (left), with grey bars indicating 95% credibility interval. Additionally, results for simulated versus measured SOC over time for different residues (right). Dots with error bars indicate the mean and 95% credibility interval of observations and simulations. The black line and grey band indicate the best simulation and the 95% credibility interval of the Bayesian calibration posterior, respectively.





**Figure 5.** Simulated versus measured values of aggregate carbon, litter carbon, mineral associated organic carbon (MAOC), microbial biomass nitrogen, soil organic carbon (SOC) and soil C/N ratio. The grey bars indicate the 95% credibility interval. The black line marks the 1 to 1 line, the blue line the regression of simulated on measured values.





**Figure 6.** Simulation of microbial nitrogen ($MIC_N$) in 2005, 2008 and 2019 (top) and of aggregate protected C ($Agg_C$; bottom left) and free mineral associated C ($MAO_C$; bottom right) of different residues in 2019. Dots with error bars indicate the mean and 95% credibility intervals of observations. The black line and grey band indicate the best simulation and the 95% credibility intervals of the Bayesian calibrations' posterior, respectively. The dashed line indicates the mean free $MAO_C$ in the Control in 2019





## 3.3 Model behavior when aggregate formation was removed

Removing the aggregate protection from the calibrated SAMM model to derive SAMMnoAGG, showed that the model assigned a high importance to aggregate protection for the process of SOC stabilization. Without aggregate protection, the simulated SOC of all treatments reduced to about half compared to measured values (Fig. 7; Table 5b). As a result, all litter addition treatments had approximately the same amount of simulated SOC (excluding litter) in SAMMnoAgg, despite their difference in C/N ratios, lignins and polyphenols (Fig. 7). Hence, removing aggregate protection led to a significantly reduced and now negative modelling efficiency (-3.68) for SOC (Table 5). In addition, the simulation of microbial nitrogen was negatively affected by removing aggregate protection. Because of the absence of aggregate protection of $LAB_C$ and $STR_C$ (i.e. POM), simulated microbial growth become too high after litter addition. However, it still had a positive modelling efficiency (reduction of EF to 0.13 from 0.24, initially) and the temporal trend of the strongest microbial growth occurring after litter addition, was still represented (simulation not shown). In contrast, removing aggregate protection had little effect on the simulation of litterbag carbon (EF was 0.79) and the increase in model error was minor because litterbag carbon is not protected by aggregates. Overll, the dipterocarp treatment was simulated to have the highest carbon storage of litter and SOC combined without aggregate protection. This was mainly because not all dipterocarp litter decomposed within one year.

## 3.4 Comparison of SAMM separately calibrated with and without the aggregate protection mechanism

When the SAMM model without aggregate formation (SAMMnoAgg) was recalibrated, the poor model performance was largely resolved (Table 5c). For example, the model performance for SOC were the same for the two models (EF of 0.68). Yet, some notable difference between SAMM and recalibrated SAMMnoAgg remained for the microbial nitrogen and litter carbon. Their dynamics were simulated slightly worse in recalibrated SAMMnoAGG compared to SAMM (EF of 0.80 versus 0.75 for litterbag C and EF of 0.24 versus 0.18 for microbial nitrogen; Table 5c). Consequently, the overall model AIC, considering, for comparability, only litterbag carbon, microbial nitrogen and SOC, was slightly lower for SAMM versus recalibrated SAMMnoAgg (5351 versus 5554).

When comparing the posterior distributions of both model versions, it became evident that the recalibration of SAMMnoAgg counteracted the loss of aggregate protection by lowering the turnover turnover of MAOM by almost an order of magnitude (about 85%; Fig. A1). This indicates that the representation of aggregate protection on SOC was changed from explicit to implicit. Also, the recalibrated SAMMnoAgg version had a lower half saturation constant for direct absorption of $LMW_{C\&N}$ to MAOM in tendency, allowing for a faster direct absorption (Table 4). Removing aggregate protection did, however, not affect most other model parameters, which were similar in their posterior distributions between SAMM and recalibrated SAMMnoAgg . Interestingly, the 95% posterior credibility intervals were smaller for SAMM than recalibrated SAMMnoAgg and at the same time covered a larger proportion of measurements of microbial nitrogen and SOC, indicating that they were more accurate for the aggregate version of SAMM.




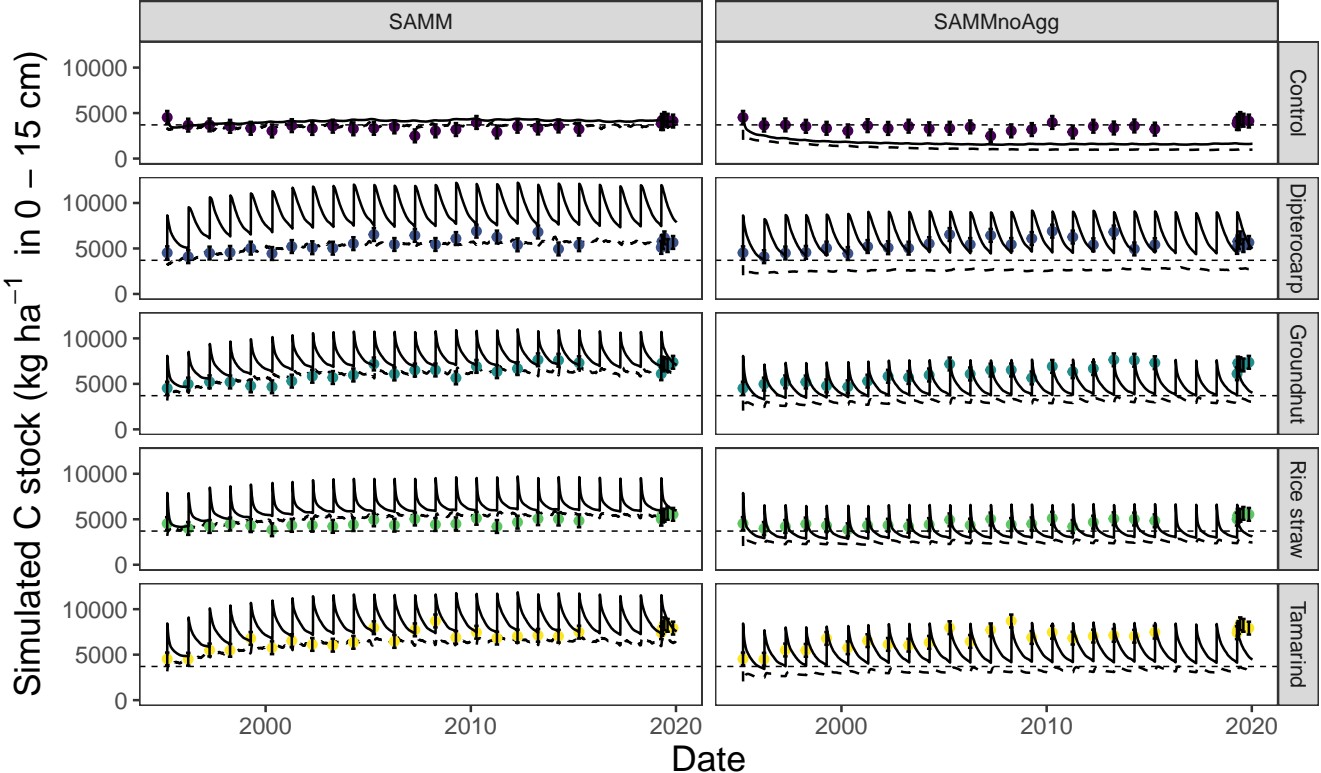

**Figure 7.** Results for the simulation of carbon stocks with the model version including aggregates (SAMM, left) and when aggregate protection is removed without recalibration (SAMMnoAgg, right). The solid line indicates all carbon including litter, the dynamic dashed line indicates the combined soil carbon stocks stored in $MAO_C$, $Agg_C$ and $MIC_C$. The horizontal dashed thin line indicates the mean measured SOC in the control. Dots with error bars indicate the mean and 95% credibility intervals of observations (excluding litter).

### 3.5 Analysis of model parameter behavior

In both calibrated model versions, SAMM showed a clear distinction between the turnover of different carbon pools (Fig. A1). The highest likelihood turnover rates of MAOM, structural and metabolic litter differed by a factor of five to ten (e.g., around 0.0004, 0.002 and 0.02 for SAMM; Table 4). The breakdown of aggregates, with around 3% per day, as well as the

physicochemical aggregate formation, equivalent to a $MIC_C$ growth of 31 kg ha$^{-1}$ per day, and were high in SAMM. This indicated a highly dynamic aggerregate fraction and a high importance assigned to physicochemical aggregate formation. At the same time, few strong parameter correlations of r > 0.4 were present in the posterior parameters set for the SAMM (Fig. A2) and the parameter correlations in recalibrated SAMMnoAgg were of similar magnitude (Fig. A3). First, the structural litter turnover and the protection capacity of structural for labile litter were correlated (r = 0.48). Then, there was a negative

correlation between aggregate protection of POM by microbial growth, and the rate of physicochemical aggregate formation




**Table 5.** Model evaluation statistics of a) the default SAMM model (with aggregate protection), b) SAMM model without aggregate protection (SAMMnoAgg), and c) recalibrated SAMM model without aggregate protection (AMMnoAgg). RMSE and the width 95% credibility intervals (w95% CI) are in kg ha$^{-1}$. Evaluation statistics are from the Bayesian calibration. EF, Nash-Sutcliffe modelling efficiency; (R)MSE, (root) mean squared error; LC, lack of correlation; NU, nonunity slope; SB, squared bias; AIC, Akaike information criterion.

| | dataset | EF | RMSE | R$^2$ | LC | NU | SB | MSE | AIC | % in 95%CI | w95% CI[a] | |
|---|---|---|---|---|---|---|---|---|---|---|---|---|
| | **a) Default SAMM model** | | | | | | | | 5351[b] | | | |
| | Litterbag C | 0.80 | 537.3 | 0.82 | 87 | 1 | 11 | 288685 | 869 | 64 | 926 | |
| | Microbial N | 0.24 | 22.8 | 0.42 | 76 | 22 | 2 | 518 | 2041 | 53 | 36 | |
| | SOC | 0.68 | 788.4 | 0.77 | 73 | 5 | 22 | 621636 | 2534 | 62 | 1381 | |
| ( | Aggregate C | 0.60 | 302.6 | 0.61 | 98 | 0 | 2 | 91548 | 521 | 93 | 1265 | ) |
| ( | Free MAO C | 0.24 | 356.4 | 0.60 | 53 | 46 | 1 | 126997 | 664 | 93 | 1188 | ) |
| ( | Soil C/N[c] | -0.58 | 6.2 | 0.04 | 61 | 35 | 4 | 38 | 1201 | 61 | 12 | ) |
| | **b) Removing aggregate protection/formation (SAMMnoAgg)** | | | | | | | | 11799[b] | | | |
| | Litterbag C | 0.79 | 540.4 | 0.81 | 89 | 1 | 10 | 291993 | 896 | | | |
| | Microbial N | 0.13 | 24.4 | 0.38 | 70 | 22 | 8 | 594 | 2183 | | | |
| | SOC | -3.68 | 2922.3 | 0.62 | 8 | 2 | 90 | 8539715 | 8855 | | | |
| | **c) Recalibrated SAMMnoAgg** | | | | | | | | 5554[b] | | | |
| | Litterbag C | 0.75 | 600.4 | 0.77 | 89 | 3 | 8 | 360447 | 993 | 64 | 953 | |
| | Microbial N | 0.18 | 23.7 | 0.39 | 75 | 25 | 0 | 563 | 2112 | 51 | 38 | |
| | SOC | 0.68 | 792.3 | 0.75 | 77 | 19 | 4 | 627769 | 2540 | 55 | 1409 | |
| ( | Soil C/N[c] | -133262 | 1791 | 0.00 | 0 | 99 | 1 | 3211117 | Inf | 65 | 41 | ) |

[a]95% witdh of the credibility interval from the Bayesian calibration posterior; [b]Overall model AIC. For comparability of model versions this was computed without Aggregate and *MAO* C and soil C/N. [c]Not used in calibration.

(r = -0.43). Also, the absorption speed of $LMW_C$ to $MOA_C$ and the turnover of MAOM were correlated (r = 0.40). Finally, the turnover of MAOM was correlated with the microbial death (r = 0.42).

# 4   Discussion

## 4.1   SAMM as a state-of-the-art soil model with measurable pools

With SAMM, we present a state-of-the-art microbe-driven coupled C/N model, suitable for field-scale application. It simulates the effect of residue stoichiometry on microbial CUE (Sinsabaugh et al., 2016) and the role of microbial growth on aggregate formation (Laub et al., 2022; Bucka et al., 2021). It contains measurable pools, is well able to simulate aggregate formation resulting from microbial growth, maintains carbon and nitrogen identity (Wang et al., 2022) inside aggregates, and it can easily be converted into a lower complexity model without aggregates (i.e., SAMMnoAgg). The model evaluation statistics (Table



5) showed that SAMM, with its representation of carbon and nitrogen in measurable pools (including litter as measurable structural and metabolic pools), is capable of capturing the relevant processes in a long-term litter addition experiment in a tropical sandy soil and handle the complexity of microbial driven aggregate formation for different litter chemical compositions. As was demonstrated, SAMM captures the differences between treatments, the temporal development of microbial biomass, and the connection between microbial growth and aggregate formation. To our knowledge, apart from an early attempt to

model in-situ aggregate stability without considering aggregate stored carbon (Abiven et al., 2008), SAMM is the first model that demonstrated this capability in a field experiment.

That the parameter correlations were low (maximum r = 0.48) compared to calibration exercises with established models such as DayCent (Necpálová et al., 2015, showed parameter correlations between turnover times of different pools of up to r = 0.9), Daisy (Laub et al., 2020, had parameter correlations between turnover of fast and slow pools of up to r = 0.8) or

ICBM (Ahrens et al., 2014, had correlations between pools up to r = 0.7), shows that the model structure of SAMM with measurable pools has a clear advantage compared to models with theory-based conceptual pools. Furthermore, that all pools can be measured facilitates calibration, as was recently shown at global scale with Millennial compared to Century (Abramoff et al., 2022). Yet, the data needed to constrain models with measurable pools at global scale may not be readily available. For example, we are not aware of other field experiments that include different litter types and follow microbial biomass, SOC and

aggregate carbon simultaneously over time. Hence, this version of SAMM was only tested at one site, and it remains to be evaluated for larger spatial scales and with a range of experiments with different quality organic amendments.

We posit that maintaining the carbon identity inside aggregates represents the next logical step for aggregate models, but are aware of the fact that the marginally better performance of SAMM vs recalibrated SAMMnoAgg only provides initial evidence. Hence, we invite others to test the concept against further data sets with SAMM or with their own model. By

maintaining the carbon identity, aggregate models can help answer important scientific questions, such as how important the stabilization of carbon in aggregates is for the global carbon cycle. As shown by disabling aggregates in SAMMnoAgg, SAMM can also provide novel insights into the relative importance of different processes, such as the importance of aggregate protection for carbon stabilization versus protection by attachment to minerals (Angst et al., 2021). In this calibration exercise, the model estimation was that only half of the carbon is protected as MAOC and that about half of the carbon is protected inside

aggregates (Fig 7). However, because we had no measurements of POM versus MOAM in aggregates, we cannot evalute this by measurements, and it is based on the assumption of complete protection of POM and MAOM inside aggregates. Another interesting process insight was that physicochemical aggregate formation was estimated by SAMM to be of similar importance as microbial aggregate formation and that both processes probably happen in parallel, especially in tropical soils as was tested here. Yet, it is clear that our data did not provide enough information to clearly distinguish between both processes, which can

be seen by the wide posterior credibility intervals of physicochemical aggregate formation. Despite this, the fact that SAMM could simulate the observed increase of aggregate C in the dry season towards the end of 2019 (Fig 6) indicates that this process needs to be included.



## 4.2   Is aggregate protection necessary to better simulate microbial and SOC dynamics?

It has been postulated that because a substantial portion of soil carbon is located within soil aggregates, soil aggregation needs
to be included into models to accurately capture reality (Segoli et al., 2013; Abramoff et al., 2018). In this paper we followed
this hypothesis and explicitly tested it by comparing the performance of SAMM with and without aggregate formation on
litter carbon, microbial nitrogen and SOC simulation across the different treatments (Table 5). Since clear connections between
microbial growth and aggregate formation have been demonstrated (Laub et al., 2022; Bucka et al., 2021; Bossuyt et al., 2001;
Denef et al., 2001), including aggregate formation in SAMM is a more realistic process representation. In alignment with
our second hypothesis, removing the soil aggregate formation did, even after recalibration of SAMMnoAgg, reduce model
performance of the non-aggregated pools, albeit not strongly. This suggests that the simulation of aggregate formation and
disruption can be useful to understand overall SOC dynamics but that SAMMnoAgg was able to artificially compensate for
the missing mechanism of aggregate protection (which, as shown by crushed aggregates incubation, e.g.,  Kpemoua et al.,
2022; Puttaso et al., 2011; Six et al., 2002, clearly exists) by reducing turnover of MAOM. What also speaks for this effect
are the smaller posterior credibility intervals of SOC, microbial nitrogen and litter carbon of the aggregate version of SAMM
compared to recalibrated SAMMnoAgg (Table 5) and that they still covered a higher percentage of observations.

   The fact that the recalibrated SAMMnoAgg model still seems to implicitly account for aggregate protection of SOC by
reducing the turnover of MAOM (Fig. A1), could suggest that aggregate formation does not need to be included into models
to accurately capture differences in SOC formation at large scales. Despite being a better process representation, limited data
availability of aggregate- and microbial dynamics may make a non-aggregate model more feasible. However, for a mechanistic
understanding, i.e., using the model as a research tool to test hypotheses, it is arguably better to include aggregate formation
and carbon protection in aggregates. In contrast, simulating aggregate protection may not be necessary to assess carbon seques-
tration potential from different management strategies. One the one hand, many processes that are relevant for soil formation
and SOC stabilization and happen inside the aggregates, may be irrelevant at the field scale (Yudina and Kuzyakov, 2019) if
they are implicitly included by adjusting other model parameters. On the other hand, we only had data to test SAMM with one
long-term experiment in one single soil type. Model parsimony and equifinality often depend on how much data is available
(Marschmann et al., 2019). Hence, it is possible that across sites, the interaction of factors such as differences in texture, litter
composition and different climates on SOC protection may be best represented by a model that includes the mechanism of
aggregate protection. For example, the improvement of the model performance of Millenial over Century also only became ev-
ident when looking at the global distribution of soil carbon (i.e., only at high latitudes is Millenial better; Abramoff et al., 2022).
Clearly, a range of field experiments that measured the temporal dynamics of aggregates together with microbial biomass and
SOC would be needed to better test and hence understand the relevance of aggregate formation to simulate SOC dynamics
across scales.



### 4.3 Potential limitations and open questions

An interesting observation is that the model assumes a rather high amount of daily carbon input through roots (about 3 kg C per ha and day for both SAMM and SAMMnoAgg) additional to the litter that is added annually through the treatment. Yet, this additional material is expected to be of a rather high C/N ratio. The parameter of daily carbon input was included for two reasons: 1) we observed weed growth in the plots, despite regular weeding, and hence assuming no additional inputs did not seem reasonable and 2) model runs with carbon inputs only from litter addition could not maintain any microbial activity in

the control, further corroborating the validity for these inputs (simulations not shown). The fact that the calibration assumed rather high root inputs is potentially due to the absence of more complex microbial traits in SAMM, such as dormancy, which some other models include (Wang et al., 2015; Blagodatsky and Richter, 1998). Further, CUE is only a function of litter C/N and not of microbial community. An earlier study showed that the different treatments led to different microbial communities (Kamolmanit et al., 2013), and communities of minimal inputs usually became more efficient at recycling carbon and nitrogen

(Dijkstra et al., 2022). The higher quality daily root carbon inputs in SAMMnoAgg compared to SAMM in that regard could be interpreted as aggregate formation within a model helping to simulate microbial biomass patterns. In fact, aggregate formation, linked to both microbial growth and physicochemical formation, was very fast. Also turnover rates were high (almost as fast as metabolic litter decomposition). This is in alignment with a recent model of aggregation at the micro scale (Zech et al., 2022). Yet, it is difficult to distinguish between the different pathways of aggregate formation. Finally, the question is to what extent

POM and MAOM are effectively protected inside aggregates. In this version of SAMM, we simulated the most extreme case of a complete protection of carbon inside aggregates, which in future versions should most likely be replaced by a decomposition reduction factor because we know that aggregates do not completely protect carbon. Yet, it will be very difficult to measure carbon turnover inside aggregates and hence to constrain such a reduction factor. Finally, a next logical step would be to include multiple soil layers into SAMM, provided a suitable water leaching function is included. The $LMW_{C/N}$ leaching to deeper soils

layers, feeding aggregate formation there should in theory help to explain SOC depth gradients.

### 5 Conclusions

We presented and evaluated the SAMM model, a state-of-the-art research model with measurable pools that can simulate the formation and turnover of aggregates under different organic amendment treatments. Overall good model evaluation statistics (EF 0.2 to 0.8, depending on observation type) and low parameter correlations (r < 0.48) suggested that the current structure of

SAMM is valuable, clearly identifiable in calibration and hence parsimonious. The results suggested that aggregate protection plays a crucial role for SOC stabilization, i.e., the model results suggested that about 50% of soil carbon was protected in aggregates, even in the sandy soil of the studied long-term experiment. While for basic research, aggregate formation should be included into models, our results indicate that with model recalibration, the absence of aggregate protection in SOM models is partly compensated by reducing turnover of the MAOM pool. Hence, if the sole goal is to represent SOM, microbial nitrogen

and litter carbon well, aggregate formation may be omitted in SOM models, especially if insufficient data on aggregates exists. It is, however, possible that this compensation within our study was only possible because the data originated from a single





site. For further evidence, studies over a range of soils and climates would be needed, which calls for more long-term studies
to include repeated measurements of aggregate and microbe dynamics.

*Code and data availability.* The full dataset used for this study, as well as the R code of SAMM version 1.0 is provided on Github via Zenodo
(https://zenodo.org/record/8086828). It may be adapted for further uses, or integrated into full ecosystem models that allow for interchanging
of the SOM part of the model.)

## Appendix A: Appendix

### A1    Detailed description of SAMM pools

#### A1.1    Structural litter pool - $STR_C$

The structural litter pool ($STR_C$) consists of lignin and polyphenols, the parts of litter which stabilize the cell wall and are
processed by microbes with a low CUE. $STR_C$ is assumed to have a carbon content of 65%, representing a lignin-typical C/H/O
ratio of 20/23/7 (Gargulak et al., 2015). Through this definition, the structural litter pool is measurable as acid detergent lignin
(Van Soest and Wine, 1968) and polyphenols (Anderson and Ingram, 1993), and it does not contain nitrogen. However, cell
walls are usually a mix of structural components with celluloses and hemicelluloses, and those do not decompose as easily as
the cell interior. This is accounted for by a simulated protection capacity of structural litter pool in the metabolic litter pool,
allowing that hemicelluloses and celluloses are protected by the presence of structural litter and their decomposition is limited
by the rate of structural litter depolimerization.

#### A1.2    Metabolic litter pool – $LAB_C$ and $LAB_N$

The metabolic litter pool contains all parts of the litter which are not part of $STR_C$. This includes cellulose, hemicellulose,
intracellular carbon and nitrogen (Campbell et al., 2016). All these components are considered to be easily available to mi-
crobial uptake if not protected by $STR_C$ and due to lower depolimerization costs, microbes usually process them with a higher
CUE. To distinguish between cell wall components and cell interior, the structural litter asserts a protective capacity on a part
of the metabolic litter. This mimics that cell wall cellulose and hemicellulose are protected by cell wall lignin. The amount of
protected metabolic carbon ($ProtLAB_{C\&N}$) is not a real pool but a linear function of carbon in the structural pool. Thus, the
cell wall components are protected by the structural components by a fixed ratio. Protected metabolic carbon is thus becoming
accessible to microbes at the same rate at which the structural pool is decomposed.

#### A1.3    Low molecular weight carbon and nitrogen pools - $LMW_C$ and $LMW_N$

The low molecular weight pool contains depolymerized carbon and nitrogen originating from all other pools and easily enters
the soil solution. All decomposed residues end up in this pool. The $LMW_{C\&N}$ pool can be measured by extraction using a $K_2SO_4$





solution. Microbes, similar to other established models, such as MEND (Wang et al., 2013) and Millenial (Abramoff et al., 2018), can consume the carbon and nitrogen in the $LMW_{C\&N}$ pool. When consumed by microbes, $LMW_C$ is subject to a variable CUE. This variable CUE is a function of the C/N ratio of $LMW_{C\&N}$, thus accounting for a C/N dependent growth respiration and spilling (Sinsabaugh et al., 2013). We used the linear function of C/N dependent CUE (Fig. A5) based on Campbell et al. (2016, equation 16B), which they based on Sinsabaugh et al. (2013). Additionally, the $LMW_{C\&N}$ pool is the only pool which

can be leached. Finally, direct adsorption of $LMW_C$ and $LMW_N$ to particles from the silt and clay fraction is possible. This was simulated using a Langmuir-type relationship such as in Wang et al. (2013), with values for this relationship estimated by Abramoff et al. (2022).

### A1.4   Microbial pools - $MIC_C$ and $MIC_N$

The $MIC_{C\&N}$ pool comprises the living soil microbial biomass that actively influences the decomposition of all other pools.

$MIC_{C\&N}$ can be measured by various techniques, such as substrate induced respiration (Kandeler et al., 1999), or the more common chloroform fumigation extraction (Vance et al., 1987), but all of these are subject to considerable uncertainty. In SAMM, the $MIC_{C\&N}$ pool actively contributes to the decomposition of other pools through a microbial activity factor ($a_{MIC}$). As the uptake of $LMW_C$ and $LMW_N$ by microbes only depends on the availability and on $a_{MIC}$, the C/N ratio of microbes is not fixed. We included indirect limits to microbial C/N through a C/N-dependent CUE and a direct limit through immobilization

of nitrogen if microbial C/N surpasses an upper boundary. A spilling of nitrogen happens for very low C/N ratios at a lower boundary: If the C/N ratio of microbes becomes smaller than a minimum C/N, the excess nitrogen is released by the microbes to avoid unrealistically low C/N ratios of the microbes (maximally half of excess nitrogen per day). Both maximum and minimum microbial C/N are calibrated parameters. The microbial pool is subject to maintenance respiration and microbial death. The carbon and nitrogen of dead microbes are split between the $LMW_{C\&N}$ and the mineral associated pool, representing soluble cell

constituents and cell wall structures, which are assumed to become directly attached to minerals (Krause et al., 2019).

### A1.5   Mineral associated organic carbon and nitrogen pools – $MAO_C$ and $MAO_N$

This pool consists of all carbon and nitrogen which is attached to silt and clay. It has been long suggested that this is the form of carbon and nitrogen with a slower average turnover than total SOM (Christensen, 2001) with a residence time of decades to millenia (Kögel-Knabner et al., 2008), even in sandy soils in the tropics (e.g. Puttaso et al., 2013). There are two ways in which

carbon and nitrogen can enter the $MAO_{C\&N}$ pools: first, microbial cell walls which attach to minerals upon microbial death and second, adsorption of $LMW_{C\&N}$. As in many models, we allow for an attachment of SOM to $MAO_{C\&N}$ in the form of microbial residues that is only limited by a partitioning constant. The adsorption of $LMW_{C\&N}$ to $MAO_{C\&N}$ on the other hand follows a Langmuir-type relationship, where the limit is determined by the amount of silt and clay in a soil (Abramoff et al., 2022). This follows recent studies that demonstrated that N-rich microbial products preferentially attach to new mineral surfaces (Kopittke

et al., 2018, 2020), while the direct sorption of $LMW_{C\&N}$ depends on the amount of fine particles (Georgiou et al., 2022).





### A1.6   Aggregate pools – $Agg_C$ and $Agg_N$

To maintain the conceptual carbon identities, the carbon and nitrogen in aggregates does not represent a single pool. Instead, the aggregates consist of part of the primary constituents $STR_C$, $LAB_{C\&N}$ and $MAO_{C\&N}$ pools, which inside aggregates are protected from decomposition ($AggSTR_C$, $AggLAB_{C\&N}$ and $AggMAO_{C\&N}$). The amounts of primary constituent entering the aggregate

protected pools at each time step are a function of microbial growth. Additionally, is also a constant physicochemical aggregate formation, representing all abiotic aggregate formation processes. While inside the aggregates there is no decomposition, a concept proposed by Luo et al. (2017) as a way to reduce the number of parameters in aggregation models.

Each carbon identity is transferred back into the pool that it originated from without any matter losses during aggregate turnover. This simple concept of protection was first proposed by Luo et al. (2017) to model aggregate protection in a par-

simonious way. In alignment with recent studies which showed that the presence of microbially-produced binding agents stabilizes aggregates (Bettermann et al., 2021; Crouzet et al., 2019), the rate of aggregate formation in SAMM is a function of microbial growth. Furthermore, SAMM allows for physicochemical aggregate formation at a constant rate (currently defined as daily microbial growth equivalent). Hence it allows for both important processes of aggregate formation; biological and physicochemical (Six et al., 2002).

### A2   Technical implementation of SAMM


The SAMM model is written in the R programming language (R Core Team, 2020), with the differential equations being solved using the deSolve package with the rk4 solver (Soetaert et al., 2010). Simulation of carbon and nitrogen dynamics are performed for the topsoil layer (0 – 15 cm). While all flows of carbon and nitrogen between pools were simulated within the SAMM model, soil water status, water leaching and temperature are external inputs, needed to drive SAMM. Measurements

of soil temperature were available from a station that is located at close distance to the experiment, and soil water content and leaching of water from the soil was simulated with HYDRUS 1D model (Šimůnek et al., 2005) based on climatic data and soil texture. Measurements done with moisture sensors during 2019 showed that the HYDRUS simulated water content matched the moisture levels and dynamical pattern of measured water content (Figure A4). To be able to calibrate SAMM to litter decomposition from a litterbag experiment, we created litterbag carbon and nitrogen pools, which was reinitialized with every

yearly litter addition and did not flow into any other pools. They decomposed at the same turnover as the normal $STR_C$ and $LAB_{C\&N}$ litter pools, but could not be protected in aggregates. Note that SOC was defined to correspond all pools combined, excluding the free $STR_C$ and $LAB_C$ pools.

### A3   SAMM model equations and additional model graphs

The following section describes the SAMM model by displaying the changes of pools (Table 3) with each time step. Inputs

into the system are only in the form of litter ($I_{STR_C}$ and $I_{LAB_C}$). The flows between pools are displayed as flows ($F_{X_1X_2}$) from the donor pool ($X_1$) to the receiving pool ($X_2$) as follows:





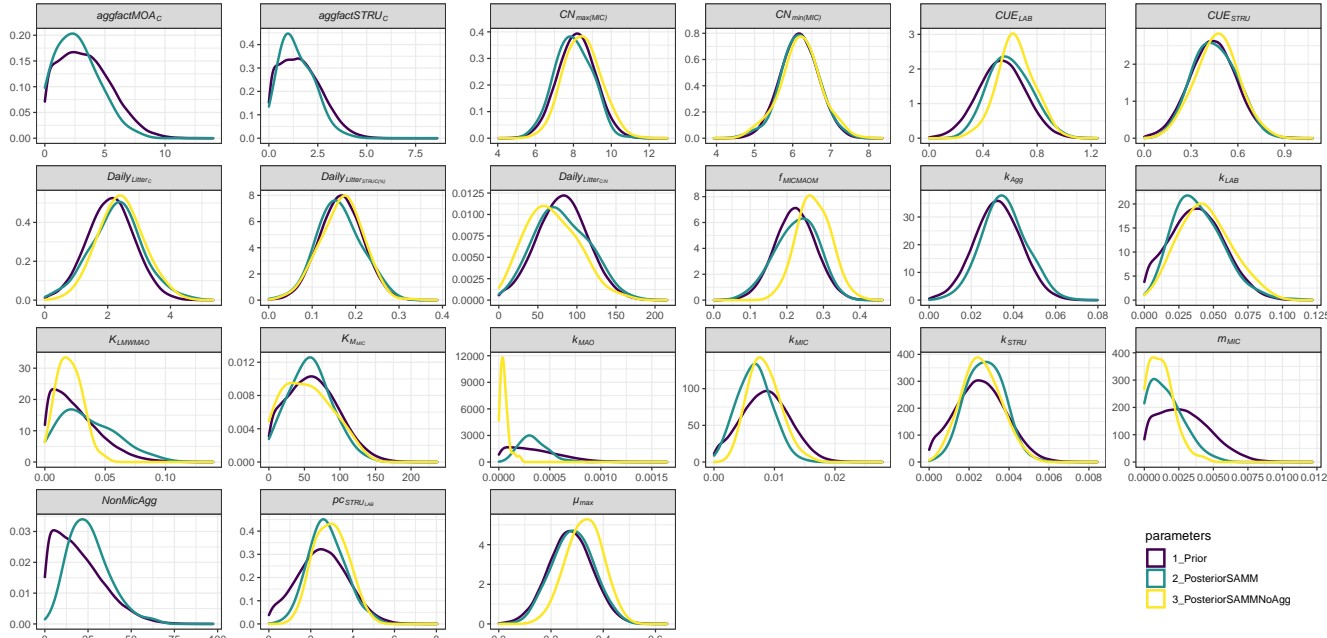

**Figure A1.** Prior and posterior parameter distributions of SAMM and the version without aggregates (SAMMnoAgg) for all model parameters that were calibrated. Priors were the mean of SAMM and SAMMnoAgg from an initial calibration of both model versions with a genetic algorithm. The width of the distribution was manually chosen and based on the range given by the genetic algorithm. Negative values were excluded.

$$\frac{dSTR_C}{dt} = +I_{STR_C} - F_{STR_C LMW_C} - F_{STR_C AggSTR_C} + F_{AggSTR_C STR_C} - F_{STR_C CO_2} \tag{A1}$$

$$\frac{dLAB_C}{dt} = +I_{LAB_C} - F_{LAB_C LMW_C} - F_{LAB_C AggLAB_C} + F_{AggLAB_C LAB_C} - F_{LAB_C CO_2} \tag{A2}$$

$$\frac{dLMW_C}{dt} = +F_{STR_C LMW_C} + F_{LAB_C LMW_C} + F_{MIC_C LMW_C} + F_{MAO_C LMW_C} - F_{LMW_C MIC_C} - F_{LMW_C MAO_C} - F_{LMW_C C_{leach}} - F_{LMW_C CO_2} \tag{A3}$$

$$\frac{dMIC_C}{dt} = +F_{LMW_C MIC_C} - F_{MIC_C LMW_C} - F_{MIC_C MAO_C} - F_{MIC_C CO_2} \tag{A4}$$

$$\frac{dMAO_C}{dt} = +F_{MIC_C MAO_C} + F_{LMW_C MAO_C} - F_{MAO_C LMW_C} - F_{MAO_C AggMAO_C} + F_{AggMAO_C MAO_C} \tag{A5}$$

$$\frac{dAggSTR_C}{dt} = +F_{STR_C AggSTR_C} - F_{AggSTR_C STR_C} \tag{A6}$$







**Figure A2.** Correlation matrix between all calibrated parameters of the SAMM model. The parameter values are from the posterior distribution of the Bayesian calibration using the SIR method.





**Figure A3.** Correlation matrix between all calibrated parameters of the model without aggregates (SAMMnoAgg). The parameter values are from the posterior distribution of the Bayesian calibration using the SIR method. Aggregate related parameters were fixed to deactivate the aggregate formation.





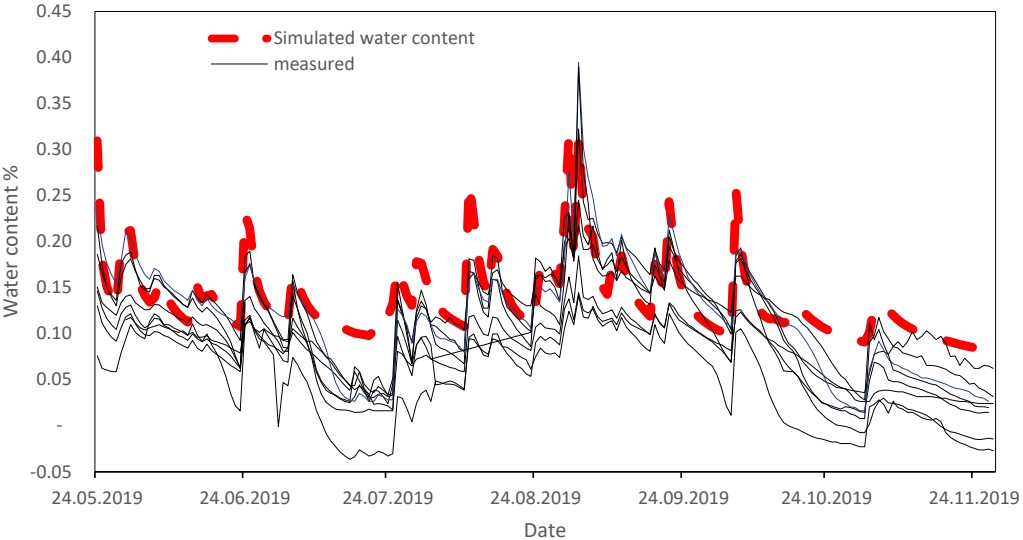

**Figure A4.** Comparison of measured water contents by moisture sensors (ECH2O EC-5, METER Group, Inc. USA; solid lines) with simulated water content by HYDRUS 1D (red dashed line). Sensors were installed in different plots of the long-term Experiment in Khon Kaen.

$$\frac{dAggLAB_C}{dt} = +F_{LAB_C AggLAB_C} - F_{AggLAB_C LAB_C} \tag{A7}$$

$$\frac{dAggMAO_C}{dt} = +F_{MAO_C AggMAO_C} - F_{AggMAO_C MAO_C} \tag{A8}$$

Respired ($CO_2$) and leached ($C_{leach}$) carbon are permanently lost from the system.

$$\frac{dCO_2}{dt} = +F_{STR_C CO_2} + F_{LAB_C CO_2} + F_{LMW_C CO_2} + F_{MIC_C CO_2} \tag{A9}$$

$$\frac{dC_{leach}}{dt} = +F_{LMW_C C_{leach}} \tag{A10}$$

The flows of carbon between pools, as described above, are computed from the state variables of each pool $X_C$, the protection capacity for the $LAB_C$ pool ($pLAB$), carbon use efficiencies for each pool ($CUE_X$) and their standard turnover rates ($k_X$) or

maximum microbial uptake for $LMW_C$ ($\mu_{max}$). Apart from $LMW_C$, the $CUE_X$, are not directly measurable, but represent a proxy for depolymerization cost. The decomposition speed of all pools outside aggregates is influenced by a reverse Michaelis-Menten microbial activity factor ($a_{MIC}$), a temperature ($s_t$) and a moisture rate modifier ($s_w$) influences all pools. Partitioning coefficients ($f_X$) are further used, where one pool feeds into several pools.

$$F_{STR_C LMW_C} = STR_C * CUE_{STR} * k_{STR} * a_{MIC} * s_t * s_w \tag{A11}$$





$$F_{STR_C CO_2} = STR_C * (1 - CUE_{STR}) * k_{STR} * a_{MIC} * s_t * s_w \tag{A12}$$

$$F_{LAB_C LMW_C} = LAB_C * (1 - pLAB) * CUE_{LAB} * k_{LAB} * a_{MIC} * s_t * s_w \tag{A13}$$

$$F_{LAB_C CO_2} = LAB_C * (1 - pLAB) * (1 - CUE_{LAB}) * k_{LAB} * a_{MIC} * s_t * s_w \tag{A14}$$

$$F_{LMW_C MIC_C} = LMW_C * CUE_{CN(LMW)} * \mu_{max} * a_{MIC} * s_t * s_w \tag{A15}$$

$$F_{LMW_C CO_2} = LMW_C * (1 - CUE_{CN(LMW)}) * \mu_{max} * a_{MIC} * s_t * s_w \tag{A16}$$

The protection and disruption of aggregates is formulated as follows:

$$F_{STR_C AggSTR_C} = min(((F_{LMW_C MIC_C} + NonMicAgg) * aggfact_{STR_C}), STR_C) \tag{A17}$$

$$F_{LAB_C AggLAB_C} = min(F_{STR_C AggSTR_C} * pc_{STR_{LAB}}, LAB_C) \tag{A18}$$

$$F_{MAO_C AggMAO_C} = min(((F_{LMW_C MIC_C} + NonMicAgg) * aggfact_{MAO_C}), MAO_C) \tag{A19}$$

$$F_{AggSTR_C STR_C} = AggSTR_C * k_{Agg} * s_t * s_w \tag{A20}$$

$$F_{AggLAB_C LAB_C} = AggLAB_C * k_{Agg} * s_t * s_w \tag{A21}$$

$$F_{AggMAO_C MOA_C} = AggMOA_C * k_{Agg} * s_t * s_w \tag{A22}$$

$$F_{MIC_C CO_2} = MIC_C * m_{mic} * s_t * s_w \tag{A23}$$

$$F_{MIC_C LMW_C} = MIC_C * k_{mic} * (1 - f_{MICMAOM}) * s_t * s_w \tag{A24}$$

$$F_{MIC_C MAO_C} = MIC_C * k_{mic} * f_{MICMAOM} * s_t * s_w \tag{A25}$$

$$F_{MAO_C LMW_C} = MAO_C * k_{MAO} * a_{MIC} * s_t * s_w \tag{A26}$$





Adsorption to MOAC is formulated as follows:

$$F_{LMW_C MAO_C} = LMW_C * K_{LMWMAO} * \frac{MAO_{C_{max}} - MAO_C}{MAO_{C_{max}}} * s_t * s_w \tag{A27}$$

For leaching, which was externally calculated using the HYDRUS 1D model (Šimůnek et al., 2005) it is assumed that
$LMW_{C\&N}$ are equally mixed with the soil solution and thus lost at the same rate as leached water.

$$F_{LMW_C C_{leach}} = min(w_{leach} * LMW_C; 0.95 * LMW_C) \tag{A28}$$

The reverse Michaelis-Menten microbial activity factor ($a_{MIC}$), which influences the decomposition speed of most pools, the
ratio of $STR_C$, $LAB_{C\&N}$ and $MAOC_{C\&N}$ protected in aggregates are calculated as follows:

$$a_{MIC} = max(\frac{MIC_C}{K_{M_{MIC}} + MIC_C}; 0.05) \tag{A29}$$

It was defined as never being lower than 0.05, so that microbes in low organic matter input treatments would not completely
die off.

The maximum adsorption capacity of a soil depends on the modeled depth, the bulk density (BD) and the amount of silt and
clay particles (SiCl):

$$M_{C_{max}} = depth * BD * \%SiCl * c_{SORP} \tag{A30}$$

The temperature ($s_t$) and a moisture scalar ($s_w$) and the dynamic CUE were adopted from established models and not subject
to further modification (Fig. A5). For the temperature scalar, an exponential equation was chosen as is common in many models
(e.g. Daisy; Hansen et al., 1993). In this context it is important to note that different temperature rate modifiers have a different
temperature at which they set the temperature scalar to 1. Here 20°C was chosen to be representative for the tropical climates.
Many temperate models use a value of 10°C for the scalar (Daisy, RothC), whereas Century and Millenial use a scalar that has
a maximum value of 1 at 40°C but only 0.5 at 20°C. This difference in temperature scalar functions needs to be considered, for
example, when adopting turnover rates from one model to another. In that case, rates need to be adjusted accordingly (e.g. in
the case of SAMM multiplying them by 2 for models that define the scalar to be 1 at 10°C and use an exponential temperature
function with a $Q_{10}$ value of 2).

$$s_t = 2^{(\frac{t-20}{10})} \tag{A31}$$

$$s_w = min\left((0.6 + 0.4 * \frac{pF}{1.5}); max(1.625 - \frac{pF}{4}; 0); 1\right) \tag{A32}$$

$$CUE_{CN(LMW)} = CUE_{LMW} * min\left(CN(LMW)^{-1} * 13.4; 1\right) \tag{A33}$$

The flow of nitrogen between the different pools is simulated in a similar way as the carbon pools:



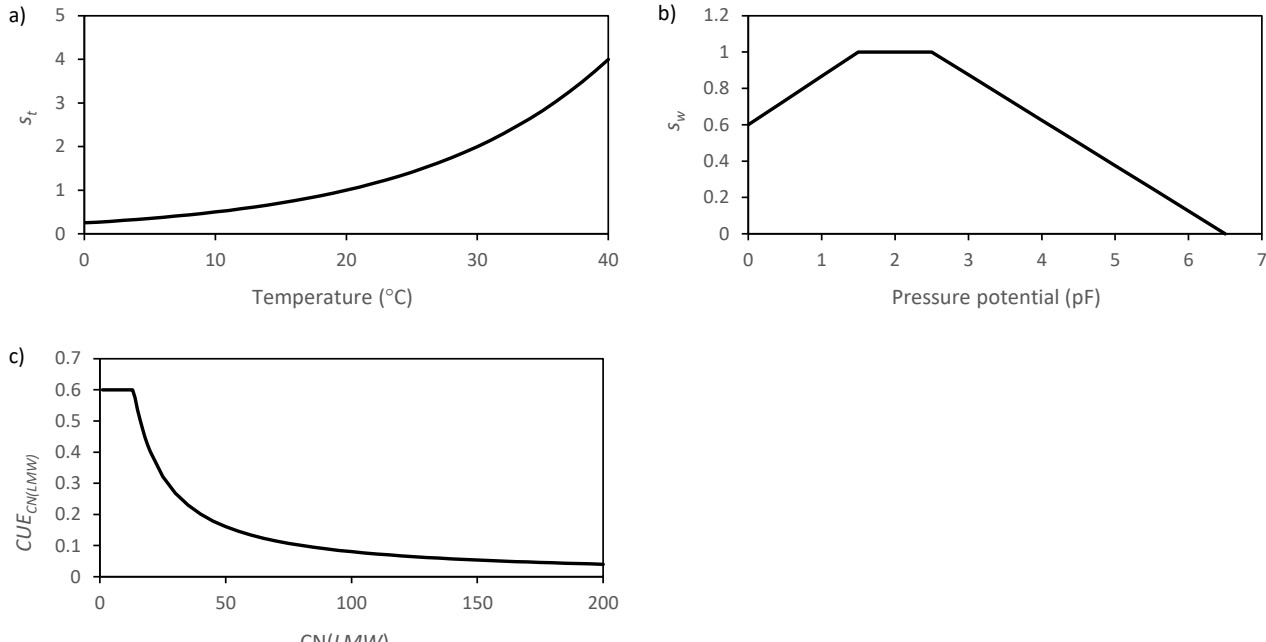

**Figure A5.** Graphic representation of the scalar functions which are applied in SAMM to represent the effect of a) temperature b) moisture. Additionally the function that represents c) the dynamic *CUE* based on the C/N ratio of $LMW_{C\&N}$ is displayed.

$$\frac{dLAB_N}{dt} = +I_{LAB_N} - F_{LAB_N LMW_N} - F_{LAB_N AggLAB_N} + F_{AggLAB_N LAB_N} \tag{A34}$$

$$\frac{dLMW_N}{dt} = +F_{LAB_N LMW_N} + F_{MIC_N LMW_N} + F_{MAO_N LMW_N} - F_{LMW_N MIC_N} - F_{LMW_N MAO_N} - F_{LMW_N N_{leach}} - IM_{MIC_N} + OS_{MIC_N} \tag{A35}$$

$$\frac{dMIC_N}{dt} = +F_{LMW_N MIC_N} - F_{MIC_N LMW_N} - F_{MIC_N MAO_N} + IM_{MIC_N} - OS_{MIC_N} \tag{A36}$$

$$\frac{dMAO_N}{dt} = +F_{MIC_N MAO_N} + F_{LMWC_N MAO_N} - F_{MAO_N LMW_N} - F_{MAO_N AggMAO_N} + F_{AggMAO_N MAO_N} \tag{A37}$$

$$\frac{dAggLAB_N}{dt} = +F_{LAB_C AggLAB_N} - F_{AggLAB_C LAB_N} \tag{A38}$$

$$\frac{dAggMAO_N}{dt} = +F_{MAO_C AggMAO_N} - F_{AggMAO_C MAO_N} \tag{A39}$$

$$\frac{dN_{leach}}{dt} = +F_{LMW_N N_{leach}} \tag{A40}$$



To calculate the flows of nitrogen, the same scalars, ratios of protected $STR_C$, $LAB_{C\&N}$ and $MAOC_{C\&N}$ in aggregates, and turnover rates are used. Additionally, the microbes can immobilize nitrogen ($IM_{MIC_N}$) from $LMW_N$, if their C/N ratio gets too wide, or spillover nitrogen to the DON pool ($OS_{MIC_N}$), if their C/N ratio gets too narrow:

$$F_{LAB_N LMW_N} = LAB_N * (1 - pLAB) * k_{LAB} * a_{MIC} * s_t * s_w \tag{A41}$$

$$F_{LMW_N MIC_N} = LMW_N * \mu_{max} * a_{MIC} * s_t * s_w + IM_{MIC_N} - OS_{MIC_N} \tag{A42}$$

$$F_{MIC_N LMW_N} = MIC_N * k_{mic} * (1 - f_{MICMAOM}) * s_t * s_w - IM_{MIC_N} + OS_{MIC_N} \tag{A43}$$

$$F_{MIC_N MAO_N} = MIC_N * k_{mic} * f_{MICMAOM} * s_t * s_w \tag{A44}$$

$$F_{MAO_N LMW_N} = MAO_N * k_{MAO} * a_{MIC} * s_t * s_w \tag{A45}$$

$$F_{LMW_N MAO_N} = F_{LMW_C MAO_C} * \frac{LMW_N}{LMW_C} \tag{A46}$$

$$F_{LAB_N AggLAB_N} = F_{LAB_C AggLAB_C} * \frac{LAB_N}{LAB_C} \tag{A47}$$

$$F_{MAO_N AggMAO_N} = F_{MAO_C AggMAO_C} * \frac{MAO_N}{MAO_C} \tag{A48}$$

$$F_{AggLAB_N LAB_N} = AggLAB_N * k_{Agg} * s_t * s_w \tag{A49}$$

$$F_{AggMAO_N MOA_N} = AggMOA_N * k_{Agg} * s_t * s_w \tag{A50}$$

$$F_{LMW_N N_{leach}} = min(w_{leach} * LMW_N; 0.95 * LMW_N) \tag{A51}$$

$$IM_{MIC_N} = if\left(\frac{MIC_C}{MIC_N} > CN_{max(MIC)}\right)\left[min\left(\frac{MIC_C}{CN_{max(MIC)}} - MIC_N; \frac{1}{2} LMW_N\right); 0\right] \tag{A52}$$

$$OS_{MIC_N} = if\left(\frac{MIC_C}{MIC_N} < CN_{min(MIC)}\right)\left[0.5\left(MIC_N - \frac{MIC_C}{CN_{min(MIC)}}\right); 0\right] \tag{A53}$$

*Author contributions.* PV and GC designed the long-term experiment and acquired funding throughout. PV maintained the experiment and supervised data generation. BK was involved in data generation for many years. SSch with help of ML generated the detailed 2019 data of aggregate dynamics. ML, GC and SB jointly developed the conceptual model, JS and MvdB gave critical feedback on it. ML developed the model equations from the conceptual model, wrote the model code and implemented the model-data fusion, MvdB helped ML in revising the model code. ML wrote the initial draft. All coauthors were involved in refining the initial draft to the submitted version.



*Competing interests.* The authors declare that they have no conflict of interest.



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
