# Peer review of "SAMM version 1.0: A numerical model for microbial mediated soil aggregate formation."

_EGUsphere, 2023_

## Author Response (AR1)

Reviewer 1:

The paper by Laub et al presents a newly developed model called SAMM. The model aims at representing the effect of aggregation on soil organic matter (SOM) dynamic. The paper is well written and the subject fits perfectly with the GMD scopes. I appreciated the approach proposed by the authors and in particular the comparison between SAMMnoagg and SAMM. It is fair to recognize that SAMMnoagg performed as well as SAMM when properly calibrated. I think this paper deserves publication after some corrections:

We thank the reviewer for the positive assessment and for the valuable suggestions to improve the manuscript.

1. The authors claimed that this is the first model considering aggregation. It is not totally true, the MIMICS model published by Wieder et al., (2014) consider a physically protected pool that could be similar to the Agg pools presented here though not as detailed as what presented by the authors.

We agree with this. Therefore, we refined the statement that "SAMM is the first model to demonstrate this capability in a field experiment with different litter qualities".

stability without considering aggregate stored carbon (Abiven et al., 2008), SAMM is the first model  to demonstrate this capability in a field experiment with different litter qualities.

2. The authors evaluated the SAMM model using microbial biomass obtained after fumigation extraction. This is problematic because the chloroform extraction method extracts the full biomass including the dormant one and, in your model, you represent the active one. You can't directly compare both since 90 % of the biomass is dormant (Lennon and Jones, 2011).

While we agree that it would be best to distinguish between active and dormant microbial biomass, this type of data was not (and is generally mostly not) available for the long-term trials and efficient measurability of all major pools was our credo. Further, SAMM links the microbial aggregate formation to microbial growth and not to microbial biomass itself (see Eqs. A17, A19 and A24-25). Therefore, absolute values of microbial biomass (estimated by CFE) should not strongly affect the simulation of aggregate dynamics. Further, the inclusion of a temperature and moisture scalar ($s_t$ and $s_w$) should be linked to that factors that control microbial dormancy. We think that in the absence of measured dormancy, this is the best alternative simulating the dynamics of active and dormant fractions of microbial biomass. For these reasons, we think that CFE measured microbial biomass is sufficient for the main purpose of this paper and model, i.e. to show that microbial dynamics play a significant role in aggregate formation.

3. The initialization procedure of the model is not detailed enough, does the simulation showed here started after a spinup? What are the consequences of the initialization procedure on the results?

To initialize the pools, we used the mean measured fractions of SOC in the rice straw treatment, which had not experienced changes in SOC over the time of the experiment. In the absence of data on historic plant input quantities and qualities, this was considered the best option. Ideally, SOC fractions at the start of any experiment would be measured.

We did a small sensitivity analysis on the influence of the initialization assumptions. For this, we perturbated the following initialization assumptions. 1) the fraction of initial litter C that was labile, 2)

how much of the initial SOC was MAOC, 3) how much of initial litter was protected in aggregates, and 4) how much of the initial MAOC was protected in aggregates (See top to lowest panel, below). We assessed the effect on SOC and on simulated C in aggregates. The perturbation was from 80% to 120% of the initial values. As can be seen, the effects are small for SOC and disappear completely after the first 5 to 10 years, and for simulated C in aggregates, they disappear after less than 3 years.

[Figure]

We added this explanation to the main text.

et al. (2022). To initialize the pools, we used the mean of the measured SOC fractions in the rice straw treatment in 2019, which had not experienced major changes in SOC since the start of the experiment. In the absence of fractionation data from the start of the experiment and historic plant input quantities and qualities, this was considered the best option. Ideally, SOC fractions would be measured at the beginning of any experiment. However, sensitivity analyses, perturbing the distribution of initial SOC between MAOC and litter pools from 80 to 120% of our initial assumptions confirmed a very limited effect (any visible differences in simulated SOC and aggregate C disappeared within less than ten and less than three years, respectively; see response to referee comments https://doi.org/10.5194/egusphere-2023-1414-AC1).

4.  I don't understand what is the rational behind the ProtLAB pools, how the presence of structural litter can protect the labile pool. It needs to be more justified.

Thanks for the hint. We realized that it was not mentioned in the main text. We added a short description in the main text and refined the description in the appendix. In short, this was a way for us to simulate soluble, non-lignin cell-wall and lignin cell-wall components with three instead of two pools.

of structural litter ($STR_C$), and the labile/metabolic litter pool ($LAB_{C\&N}$). To distinguish between the cell walls and the interior part of $LAB_{C\&N}$, the $STR_C$ protects part of $LAB_{C\&N}$ from decomposition ($ProtLAB$ pool; see Appendix A1), mimicking that part of $LAB_{C\&N}$ is interviewed with $STR_C$ in the cell walls. Upon depolymerization, the carbon and nitrogen of any pool

5.  It is not clear what the time step of the model is, please clarify.

Thanks for pointing this out. We added the clarification that the model is coded in R and run with the deSolve package. It can thus be run at any time step. We used a daily time step with the optimized rk4() solver, after confirming that the results for this were the same than using an ode() solver, which makes time steps infinitely small and has no numerical errors.

The SAMM model  was written in the R programming language (R Core Team, 2020),  and differential equations were solved using the deSolve package with the rk4 solver (Soetaert et al., 2010).  Thus, it can be run at any time step. We used a daily time step with the optimized rk4() solver, after confirming that the results for this were the same as using an ode() solver, which makes time steps infinitely small and has no numerical errors. Simulations

6.  In the main text, the information on the boundary's conditions is not clear. A sentence refereeing the appendix would help the reader to find the information.

We were not 100% sure what boundary's conditions you referred to. We interpreted this statement in the context of the upper and lower boundary for MIC pool`s C:N ratio. We thus added the following sentence" Further, MICC&N can immobilize or release N, to maintain their C:N ratio (see appendix A1.4)."

back into the $LMW_{C\&N}$ pool. Furthermore, $MIC_{C\&N}$ can immobilize or release N, to maintain their C/N ratio (see Appendix A1.4). Direct adsorption of $LMW_{C\&N}$ to $MAO_{C\&N}$ is also possible. Carbon and nitrogen from the primary constituents (i.e.,

7.  Since the model is newly developed a mass balance calculation showing that the mass balance is closed is necessary to trust the model behavior.

We agree and in fact we have included this, we just did not mention it in the text (See the model code that we published on Zenodo contains a mass balance equation, causing the model to stop if the mass balance is not closed). We now made this explicit "we added a mass balance equation to stop the model with an error message, if the mass balance is not closed."

details Details are described in appendix the Appendix A2. As SAMM is a new model, we added a mass balance equation to stop the model with an error message if the mass balance was not closed. Further, most model parameters needed to be

8. From Fig. 7 it is not clear whether the prediction of SAMMnoAgg recalibrated are different from SAMM. It might be interesting to test through a statistical analysis if the two models give predictions that are significantly different.

As you correctly pointed out, it cannot be seen in Fig. 7, but the evaluations statistics of Table 5 show that overall SAMM performs significantly better than SAMMnoAgg for the joint evaluation of litterbag C, microbial N and SOC. For SOC, there is almost no difference, though.

9. L46-49. You should write "One of the important processes…" not only CUE matters

We agree that CUE is one of the most important processes and therefore added "It is considered a key factor I stabilizing SOC (Cotrufo et al., 2013)." to the sentence.

(Sinsabaugh et al., 2013) have on the carbon use efficiency (CUE) of microbes. the microbes. It is considered a key factor in stabilizing SOC (Cotrufo et al., 2013). For example, Lavallee et al. (2018) showed that shoot material leads to more stabilized

10. L95: The data should be show in supp mat because you may have no significant changes for 2 mains reasons :1. There is indeed no or a very limited effect or 2. The variance between plots is so high that the statistical power of your setup is not strong enough to detect any change.

Thanks for this suggestion. We now display the bulk density data for the years where it was available in the supplement (see below). The only effect was difference between different sampling years. The most realistic explanation for this was that differences were caused by varying personal conducting the sampling. Given the high sand content of the site, we are also not surprised to only find a limited effect on soil structure. Since we used a mixed model with a nested random effect, random variability in the field based on position should be accounted for and we think the reason is the first that you specified.

[Figure]

**Figure A5.** Comparison of measured bulk densities in 0-15 cm in the years with available data. Treatment differences were not significant but a significant effect of year existed. This was however not considered to be any temporal trend but rather an effect arising from different people conducting the sampling. All statistical test conducted with a mixed linear effects model, containing a random intercept per subplot nested in the experimental block.

11.  L307: This comparison is not totally fair because you are comparing with 1st order kinetics models, you should compare with Millenial and MIMICS.

We agree that this would be better, but we did not find any info on the parameter correlation of these models. We thus added "It could, however, also be due to the superiority of Michaelis Menten to first order kinetics."

with theory-based conceptual pools.  It could, however, also be due to the superiority of Michaelis Menten to first-order kinetics. Furthermore, the fact that all pools can be measured facilitates calibration, as was recently shown  on a

References cited

Lennon, J. T. and Jones, S. E.: Microbial seed banks: the ecological and evolutionary implications of dormancy., Nat. Rev. Microbiol., 9, 119–130, https://doi.org/10.1038/nrmicro2504, 2011.

Wieder, W. R., Grandy, A. S., Kallenbach, C. M., and Bonan, G. B.: Integrating microbial physiology and physio-chemical principles in soils with the MIcrobial-MIneral Carbon Stabilization (MIMICS) model, Biogeosciences, 11, 3899–3917, https://doi.org/10.5194/bg-11-3899-2014, 2014.

Reviewer 2:

The manuscript by Laub et al. presents a new, advanced soil carbon dynamics model featuring a measurable-pools structure, which includes an explicit aggregate formation process and its connection with microbial growth. The model parameters were calibrated against measurements in a long-term experiment at a tropical site, showing low parameter correlations that indicate a parsimonious model structure. Their model results could reasonably reproduce the observed microbial biomass and soil carbon changes after litter addition, and highlighted the role of aggregate protection which accounted for about half of soil carbon stabilization at the tested site. Overall, the manuscript is well written and logically organized. Model limitations are also well discussed. I have only some minor comments that need to be addressed/clarified.

Thanks a lot for this overall very positive assessment.

For the Bayesian calibration of the parameters, it is not clear what data from the observations were used for the optimization. Do you use all the observations including the time series of carbon changes for different pools after each litter addition? If so, the model evaluation metrics actually represent the potentially highest level that the model can reach, which would be expected to degrade when applied to other sites.

Thanks for pointing this out. We now specify that "To calibrate SAMM and SAMMnoAgg, we used all available data of litterbag C, microbial N, SOC, while data of aggregate C and free MAOC was only used to calibrate SAMM." We further added the following sentence to the discussion "It is likely that across a range of sites, SAMM model performance will be lower, and that the calibration to the single site of this study may have resulted in an overfitting of some parameters."

scales and with a range of experiments with different quality organic amendments. It is likely that across a range of sites, SAMM model performance will be lower and that the calibration to the single site of this study resulted in an overfitting of some parameters.

It is not clear how the initial state of the model was derived. Was a spin-up process employed to reach equilibrium, or were initial values prescribed for each pool?

To initialize the pools, we used the mean measured fractions of SOC in the rice straw treatment, which had not experienced changes in SOC over the time of the experiment. In the absence of data on historic plant input quantities and qualities, this was considered the best option. Ideally, SOC fractions at the start of any experiment would be measured. See answer 3 to reviewer 1 for more details.

et al. (2022). To initialize the pools, we used the mean of the measured SOC fractions in the rice straw treatment in 2019, which had not experienced major changes in SOC since the start of the experiment. In the absence of fractionation data from the start of the experiment and historic plant input quantities and qualities, this was considered the best option. Ideally, SOC fractions would be measured at the beginning of any experiment. However, sensitivity analyses, perturbing the distribution of initial SOC between MAOC and litter pools from 80 to 120% of our initial assumptions confirmed a very limited effect (any visible differences in simulated SOC and aggregate C disappeared within less than ten and less than three years, respectively; see response to referee comments https://doi.org/10.5194/egusphere-2023-1414-AC1).

Line 222: Is there an explanation for the 1~2 months delay in the peak of MICc compared to the peak of LMWc?

Most likely this is related to a slower death rate of microbes than uptake of LMWc. The peak of MICc growth is at the exact same time as the peak in LMWc.

Figure 2: It would be helpful to add a panel showing changes of the total SOC. Besides, line colors for the different pools are a bit difficult to distinguish, please consider using more distinct colors. "STRUc" in the legend should be "STRc".

Thanks - we have included SOC into this figure, now and changed to STRc. The choice of the color scheme (Magma of viridis package; https://cran.r-project.org/web/packages/viridis/vignettes/intro-to-viridis.html) was a decision to adhere to the colorblind-friendly requirement of GMD. It is purposefully aligned with the colors of Figure 2 and, based on different options we compared, the best option to display that many pools in a colorblind-friendly way.

[Figure]

**Figure 2.** Exemplary SAMM model behavior  _and_ carbon _pool_ dynamics of the groundnut treatment in the year 2001 to 2002 starting a day before the addition of litter. The top figure displays all carbon pools inside and outside of aggregates, while the bottom figure displays the soil water content (model driver, simulated by HYDRUS 1D). In the two figure, aggregate protected pools (Agg) are represented by a dashed line, decomposable (Free) pools by a solid line. $STR_C$, structural litter; $LAB_C$, labile litter; $LMW_C$, low molecular weight; $MIC_C$, microbial; $MAO_C$, _mineral-associated_.

There are a few typos in the current manuscript, such as "MAOc" being written as "MOAc" in some places, "depolimerization", "One the one hand". Please check carefully throughout the text.

Thanks for the hint – we have corrected the mentioned mistakes and went carefully through the whole manuscript again with this in mind. We also applied the AI tool Writeful to detect and eliminate further typos.

The current abstract is not a very concise and engaging summary of the study, please refine it.

We rewrote major parts of the abstract with the goal of refining it. We hope that you consider the current form of the abstract (see below) to be more concise and engaging. We also added the track-changed version.

Maintaining soil organic matter (SOM) is crucial for healthy and productive agricultural soils and requires understanding at the process level, including the role of SOM protection by soil aggregates and the connection between microbial growth and aggregate formation. We developed the Soil Aggregation through Microbial Mediation (SAMM) model, to represent this important connection. The pools of SAMM are fully measurable, and we calibrated and evaluated it against data from a long-term bare-fallow experiment in a tropical sandy soil. This experiment received additions of plant litter of different compositions, which resulted in twice the soil carbon stocks in the best treatment compared to the control (about 8 vs. 4 t C ha$^{-1}$ in 0-15cm soil depth) after 25 years. As hypothesized, the SAMM model effectively represented the microbial growth response after the addition of litter and the subsequent formation and later destabilization of aggregates. The low correlations between different calibrated model parameters (r < 0.5 for all parameters; r > 0.4 for only 4 of 22) showed that SAMM is parsimonious. SAMM was able to capture differences between treatments in soil organic carbon (Nash-Sutcliffe modeling efficiency (EF) of 0.68), microbial nitrogen (EF of 0.24) and litter carbon (EF of 0.80). The amount of carbon within the aggregates (EF of 0.60) and in the free silt and clay fraction (EF of 0.24) was also simulated very well to satisfactory. Our model results suggested that in spite of the sandy soil, up to 50% of carbon stocks were stabilized through aggregate protection mechanisms; and that microbial and physical aggregate formation coexist. A version of the SAMM model without aggregate protection (SAMMnoAgg) initially failed to stabilize soil organic carbon (EF decreased to -3.68) and the simulation of microbial nitrogen worsened (EF of 0.13). By recalibrating SAMMnoAgg, it was possible to partially correct for the lack of aggregate protection by reducing the rate of mineral-attached carbon decomposition by about 85% (EF of 0.68, 0.75 and 0.18 for SOC, litter carbon and microbial nitrogen, respectively). However, the slightly better evaluation statistics of SAMM (e.g., Akaike information criterion of 5351 vs. 5554) suggest that representing aggregate dynamics within SOM models can be beneficial and necessary to understand the mechanism behind SOM dynamics. Our results indicate that current models without aggregate formation partly compensate for the absence of aggregate protection by lowering the turnover rates of other pools. Thus, they remain suitable options where data on aggregate associated carbon are not available.

**Abstract.**  Maintaining soil organic matter (SOM)  is crucial for healthy and productive agricultural soils  and requires understanding at the process level, including the role of .  SOM protection by soil aggregates and the connection between microbial growth and aggregate formation.  We developed the Soil Aggregation through Microbial Mediation (SAMM)  model, to represent this important connection. The pools of SAMM are fully measurable, and we calibrated and evaluated it against data from a  long-term bare-fallow experiment in a tropical sandy soil. This experiment received additions of plant litter of different compositions, which resulted in twice the soil carbon stocks in the best treatment compared to the control (about 8 vs. 4 t C ha$^{-1}$ in 0-15cm soil depth) after 25 years.  As hypothesized, the SAMM model effectively represented the microbial growth response after  the addition of litter and the subsequent formation and later  destabilization of aggregates.  The low correlations between different calibrated model parameters (r < 0.5 for all parameters; r > 0.4 for only 4 of 22 ) showed that SAMM is  parsimonious. SAMM was able to capture differences between treatments in soil organic carbon (Nash-Sutcliffe  modeling efficiency (EF) of 0.68), microbial nitrogen (EF of 0.24) and litter carbon (EF of 0.80).  The amount of carbon within the aggregates (EF of 0.60) and  in the free silt and clay fraction (EF of 0.24)  was also simulated very well to satisfactory.  Our model results suggested that  up to 50% of carbon  stocks were stabilized through aggregate protection  mechanisms; and that microbial and physical aggregate formation coexist.  A version of the SAMM model without aggregate protection (SAMMnoAgg) initially failed to stabilize soil organic carbon (EF  decreased to -3.68) and  the simulation of microbial nitrogen worsened (EF of 0.13). By  recalibrating SAMMnoAgg, it was possible to  partially correct for the lack of aggregate protection by reducing the  rate of mineral-attached carbon decomposition by about 85% (EF of 0.68, 0.75 and 0.18 for SOC, litter carbon and microbial nitrogen, respectively).  However, the slightly better evaluation statistics of SAMM (e.g., Akaike information  criterion of 5351 vs. 5554)  suggest that representing aggregate dynamics within SOM models can be beneficial and necessary to understand the mechanism behind SOM dynamics. Our results indicate that current models without aggregate formation partly compensate for the absence of aggregate protection by lowering the  turnover rates of other pools. Thus, they remain suitable options where data on aggregate associated carbon  are not available.

---

## Author Response (AR2)

**Editor:**

Dear authors,

thank you for submitting a revised version of your manuscript, which has now been seen again by the two reviewers. As you see, referee 1 still expresses concern about the modelling and benchmarking of microbial biomass. Please address this comment. I have also noticed that Table 5 contains brackets that are not documented in some rows. Please add the required information to the table caption. I look forward to receiving your revised mansucript.

Sincerely,

Chris Folberth

Dear Chris Folberth,

Thank you for the overall positive assessment and for spotting the omission in Table 5. We have added the clarification: "Data rows in brackets were not used in the calculation of the overall model AIC." We also address the concern of reviewer 1 below and added further text to highlight the issue.

We feel strongly that a measure of microbial biomass must be included in SAMM to adhere to our goal of being a measurable model with structural identity. In our opinion, taking out this measure would invalidate the main goal of the paper to connect microbial growth dynamics with aggregate formation in a measurable way. In the absence of data on dormancy, we felt that the best way to address this was in stating the limitation (that ideally, dormancy would be included) in the discussion. We hope that these modifications are acceptable to the reviewers and you.

Thanks a lot in advance,

The authors

**Reviewer1:**

The revised version of the manuscript submitted by Laub et al is much better and clearer than the previous version.

I nevertheless still have one issue on the microbial biomass. The authors acknowledge that comparing the biomass of the model with the CFE data is not ideal. I understood their arguments that the total biomass is not directly linked with aggregate dynamic and I agree with them but I would suggest to remove the biomass comparison in Fig 5, 6 and Table 5 since CFE data are not directly comparable with the model outputs.

Thank you for the overall positive assessment. We can understand your concern and agree that ideally, we would include a pool of dormant microbes and a pool of active microbes as in the MEND model (Wang et al., 2015). However, our credo was full measurability of pools and we simply had no measurements of microbial dormancy because it is actually very difficult to assess with any available measurement techniques (i.e. the closest estimate is with RNA based methods that are not straightforward in soils). Furthermore, microbial metabolic states are a continuum and defining dormancy in a binary way (dormant vs active) is thus difficult and a large oversimplification of reality (McDonald et al, 2023). It would thus add complexity without a measurable counterpart. As you stated earlier, CFE microbial biomass is representative of all alive microbes, whether dormant or not, and our single MIC pool thus comprises the full CFE microbial biomass, since all $LMW_{C/N}$ that is taken up by microbes and that is not respired ends up in the MIC pool. Hence, we think that the presentation of model outputs together with measured microbial biomass in Fig 5, 6 and Table 5 is appropriate in our case where total microbial biomass was measured and modelled. We agree that the microbial activity factor ($a_{MIC}$) should ideally be a function of active microbes only and thus we added a section to the limitations to address this issue (as shown below). Since the half saturation constant of the reverse Michaelis Menten kinetics that influences $a_{MIC}$ is a calibrated parameter ($K_{MMIC}$), we do not see a major issue for model behavior due to this discrepancy. However, we acknowledge that in cases where data on microbial dormancy would be available, it would be worth to consider two MIC pools. It is also reasonable to believe that for field scale modeling variable model structures (with or without splitting of microbial biomass into active and dormant fraction) could successfully describe SOM dynamics (see e.g. Sulman et al., 2018).

some other models include (Wang et al., 2015; Blagodatsky and Richter, 1998). In fact, the estimation of microbial biomass via chloroform fumigation extraction does not separate between dormant and active microbes. While Wang et al. (2015) suggested

390 that a model that includes dormancy can better represent the total magnitude of microbial biomass, an important concept of SAMM was the measurability of all pools and the inclusion of dormancy would thus need data on dormancy, which was not available in this trial. Ideally, only the non-dormant microbes should be considered in the microbial activity factor ($a_{MIC}$). Since $K_{M_{MIC}}$, which determines $a_{MIC}$, is a calibrated parameter, this discrepancy does not drastically alter model behavior. However,

it means that the microbial activity factor of SAMM cannot be directly compared to measurements of microbial activity, and it

395 implicitly assumes that the fraction of dormant microbes is constant. Since the water and temperature rate modifiers indirectly account for differences in microbial activity between optimal and poor conditions, the use of chloroform fumigation extraction data is most likely, in the absence of data on dormancy, the best way to represent a single microbial biomass pool while maintaining structural identity.

**References**

McDonald, M.D., Owusu-Ansah, C., Ellenbogen, J.B., Malone, Z.D., Ricketts, M.P., Frolking, S.E., Ernakovich, J.G., Ibba, M., Bagby, S.C., Weissman, J.L., 2023. What is microbial dormancy? Trends in Microbiology. https://doi.org/10.1016/j.tim.2023.08.006

Sulman, B.N., Moore, J.A.M., Abramoff, R., Averill, C., Kivlin, S., Georgiou, K., Sridhar, B., Hartman, M.D., Wang, G., Wieder, W.R., Bradford, M.A., Luo, Y., Mayes, M.A., Morrison, E., Riley, W.J., Salazar, A., Schimel, J.P., Tang, J., Classen, A.T., 2018. Multiple models and experiments underscore large uncertainty in soil carbon dynamics. Biogeochemistry. https://doi.org/10.1007/s10533-018-0509-z

Wang, G., Jagadamma, S., Mayes, M.A., Schadt, C.W., Megan Steinweg, J., Gu, L., Post, W.M., 2015. Microbial dormancy improves development and experimental validation of ecosystem model. The ISME Journal 9, 226–237. https://doi.org/10.1038/ismej.2014.120